# MonoSDF: Exploring Monocular Geometric Cues for Neural Implicit Surface Reconstruction

Zehao Yu[1]    Songyou Peng[2,3]    Michael Niemeyer[1,3]    Torsten Sattler[4]    Andreas Geiger[1,3]

[1]University of Tübingen    [2]ETH Zurich    [3]MPI for Intelligent Systems, Tübingen
[4]Czech Technical University in Prague

https://niujinshuchong.github.io/monosdf

## Abstract

In recent years, neural implicit surface reconstruction methods have become popular for multi-view 3D reconstruction. In contrast to traditional multi-view stereo methods, these approaches tend to produce smoother and more complete reconstructions due to the inductive smoothness bias of neural networks. State-of-the-art neural implicit methods allow for high-quality reconstructions of simple scenes from many input views. Yet, their performance drops significantly for larger and more complex scenes and scenes captured from sparse viewpoints. This is caused primarily by the inherent ambiguity in the RGB reconstruction loss that does not provide enough constraints, in particular in less-observed and textureless areas. Motivated by recent advances in the area of monocular geometry prediction, we systematically explore the utility these cues provide for improving neural implicit surface reconstruction. We demonstrate that depth and normal cues, predicted by general-purpose monocular estimators, significantly improve reconstruction quality and optimization time. Further, we analyse and investigate multiple design choices for representing neural implicit surfaces, ranging from monolithic MLP models over single-grid to multi-resolution grid representations. We observe that geometric monocular priors improve performance both for small-scale single-object as well as large-scale multi-object scenes, independent of the choice of representation.

## 1   Introduction

3D reconstruction from multiple RGB images is a fundamental problem in computer vision with various applications in robotics, graphics, animation, virtual reality, and more. Recently, coordinate-based neural networks have emerged as a powerful tool for representing 3D geometry and appearance. The key idea is to use compact, memory efficient multi-layer perceptrons (MLPs) to parameterize implicit shape representations such as occupancy or signed distance fields. While early works [8, 37, 44] relied on 3D supervision, several recent works [41, 60, 75] use differentiable surface rendering to reconstruct scenes from multi-view images. At the same time, neural radiance fields (NeRFs) [38] achieved impressive novel view synthesis results with volume rendering techniques. [43, 69, 74] combine surface and volume rendering for the task of 3D reconstruction by expressing volume density as a function of the underlying 3D surface, which in turn improves scene geometry.

Current neural implicit-based surface reconstruction approaches achieve impressive reconstruction results for simple scenes with dense viewpoint sampling. Yet, as shown in the first row of Fig. 1, they struggle in the presence of limited input views (DTU with 3 views) or for scenes that contain large textureless regions (walls in ScanNet or Tanks & Temples). A key reason for this behavior is that these model are optimized using a per-pixel RGB reconstruction loss. Using only RGB images as input leads to an underconstrained problem as there exist an infinite number of photo-consistent

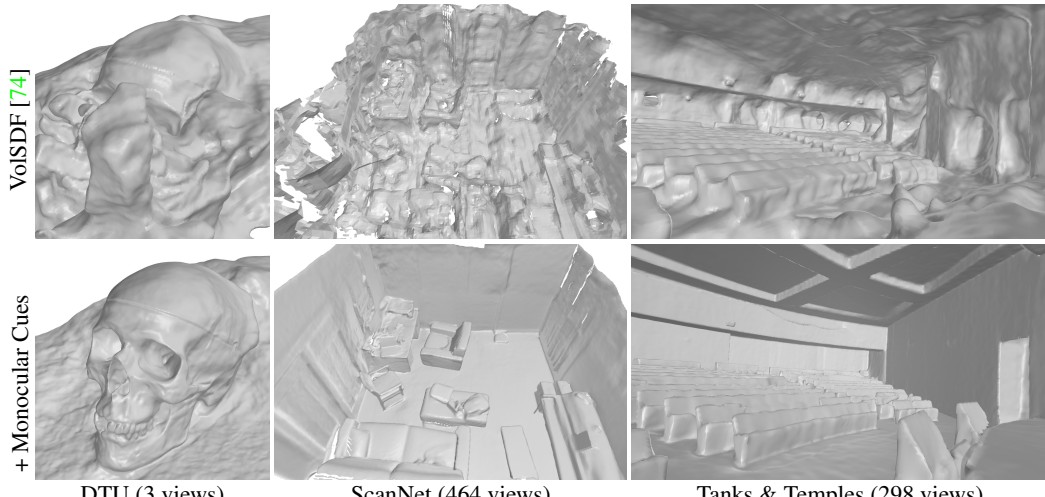

Figure 1: **MonoSDF.** Top: State-of-the-art neural implicit surface reconstruction methods fail in the presence of limited input views or when applied to complex multi-object scenes. Bottom: We demonstrate that incorporating geometric cues from general-purpose monocular predictors enables scaling to larger scenes while yielding more accurate reconstructions and speeding up optimization. An image resolution of $384 \times 384$ pixels was used for all results shown above.

explanations [4, 80]. Previous works address this problem by incorporating priors on the structure of the scene into the optimization process, e.g., depth smoothness [40], surface smoothness [43, 81], semantic similarity [27], or Manhattan world assumptions [18]. In this paper, we explore monocular geometric priors as they are readily available and efficient to compute. We show that using such priors significantly improves 3D reconstruction quality in challenging scenarios (see second row of Fig. 1).

Estimating geometric cues such as depth and normals from a single image has been an active research area for decades. The seminal work by Eigen et al. [15, 16] showed that learned models based on deep convolutional neural networks (CNNs) significantly improved over early work in this area [21–24, 53–55]. Recent work [14, 49, 50], in particular Omnidata [14], has made significant headway in terms of prediction quality and generalization to new scenes using very large datasets for training. These strong results on individual images, and the fact that monocular geometric cues can be computed efficiently, naturally lead to the question whether such models are able to provide the additional constraints required by implicit neural surface reconstruction approaches to handle more challenging settings.

This paper describes a framework, called MonoSDF, for integrating monocular geometric priors into neural implicit surface reconstruction methods: given multi-view images, we infer depth and surface normals for each image, and use them as additional supervision signals during optimization together with the RGB image reconstruction loss. We observe that these priors lead to significant gains in reconstruction quality, especially in textureless and less-observed areas as shown in Fig. 1. This is due to the fact that the photometric consistency cues used by surface reconstruction methods and the recognition cues used by monocular networks are complementary: while photometric consistency fails in textureless regions such as walls, surface normals can be predicted reliably in these areas due to the structured 3D scene layout. Conversely, photoconsistency cues allow for establishing globally accurate 3D geometry in textured regions, while normal and (relative) depth cues only provide local geometric information.

Apart from incorporating monocular geometric cues, we provide a systematic study and analysis of state-of-the-art design choices for coordinate-based neural representations in the context of implicit surface reconstruction. More specifically, we investigate the following architectures: a single, large MLP [43, 69, 74, 75], a dense SDF grid [28], a single feature grid [25, 33, 47, 48] and multi-resolution feature grids [9, 19, 39, 63, 82]. We observe that MLPs act globally and exhibit an inductive smoothness bias while being computationally expensive to optimize and evaluate. In contrast, grid-based representations benefit from locality during optimization and evaluation, hence

they are computationally more efficient. However, reconstructions are noisier for sparse views or less-observed areas. Including monocular geometric priors improves neural implicit reconstruction results across different settings with faster convergence times and independent of the underlying representation.

In summary, we make the following contributions:

- We introduce *MonoSDF*, a novel framework which exploits monocular geometric cues to improve multi-view 3D reconstruction quality, efficiency, and scalability for neural implicit surface models.

- We provide a systematic comparison and detailed analysis of design choices of neural implicit surface representations, including vanilla MLP and grid-based approaches.

- We conduct extensive experiments on multiple challenging datasets, ranging from object-level reconstruction on the DTU dataset [1], over room-level reconstruction on Replica [61] and Scan-Net [12], to large-scale indoor scene reconstruction on Tanks and Temples [30].

## 2 Related Work

**Architectures for Neural Implicit Scene Representations.** Neural implicit scene representations or neural fields [71] have recently gained popularity for representing 3D geometry due to their expressiveness and low memory footprint. Seminal works [8, 37, 44] use a single MLP as the scene representation and show impressive object-level reconstruction quality, but they do not scale to more complicated or large-scale scenes due to the limited model capacity. Follow-up works [9, 19, 36, 39, 48, 63, 82] combine an MLP decoder with one or multi-level voxel grids of low-dimensional features. Such hybrid representations are able to better represent fine geometric details and can be evaluated fast. However, they lead to a larger memory footprint with increasing scene size. In this paper we provide a systematic comparison of four architectural design choices for *implicit surface reconstruction*.

**3D Reconstruction from Multi-view Images.** Reconstructing the underlying 3D geometry from multi-view images is a long-standing goal of computer vision. Classic multi-view stereo (MVS) methods [2, 5–7, 31, 31, 56, 58, 59] consider either feature matching for depth estimation [5, 56] or represent shapes with voxels [2, 6, 7, 31, 45, 58, 65, 66]. Learning-based MVS methods usually replace some parts of the classic MVS pipeline, e.g., feature matching [20, 32, 35, 67, 79], depth fusion [13, 51], or inferring depth from multi-view images [26, 72, 73, 77]. In contrast to the explicit scene representations used by classic MVS algorithms, recent neural approaches [34, 42, 75] represent surfaces via a single MLP with continuous outputs. Learned purely from posed 2D images, they show appealing reconstruction results and do not suffer from discretization. However, accurate object masks are required. Inspired by the density-based volume rendering in NeRF [38], which demonstrated impressive view synthesis without object masks, several works [43, 69, 74] use volume rendering for neural implicit surface reconstruction without masks. However, these methods lead to poor results in large-scale scenes with textureless regions. In this work, we show that incorporating monocular priors allows these approaches to obtain significantly more detailed reconstructions and to scale to larger and more challenging scenes.

**Incorporating Priors into Neural Scene Representations.** Several researchers proposed to incorporate priors such as depth smoothness [40], semantic similarity [27], or sparse MVS point clouds [52] for the task of *novel view synthesis* from sparse inputs. In contrast, in this work, our focus is on implicit 3D surface reconstruction. Concurrently, Manhattan-SDF [18] uses dense MVS depth maps from COLMAP [57] as supervision and adopts Manhattan world priors [10] to handle low-textured planar regions corresponding to walls, floors, etc. Our approach is based on the observation that data-driven monocular depth and normal predictions [14] provide high-quality priors for the full scene. Incorporating these priors into the optimization of neural implicit surfaces not only removes the Manhattan world assumption [10] but also results in improved reconstruction quality and a simpler pipeline.[1] Compared to NeuRIS [68], a concurrent work that proposes to use normal priors for indoor scene reconstruction, we integrate monocular depth cues and further demonstrate the effectiveness of monocular cues on various neural scene representations, ranging from MLP to multi-resolution feature grids.

---

[1]Manhattan-SDF [18] requires semantic segmentation to determine where to enforce the assumption.

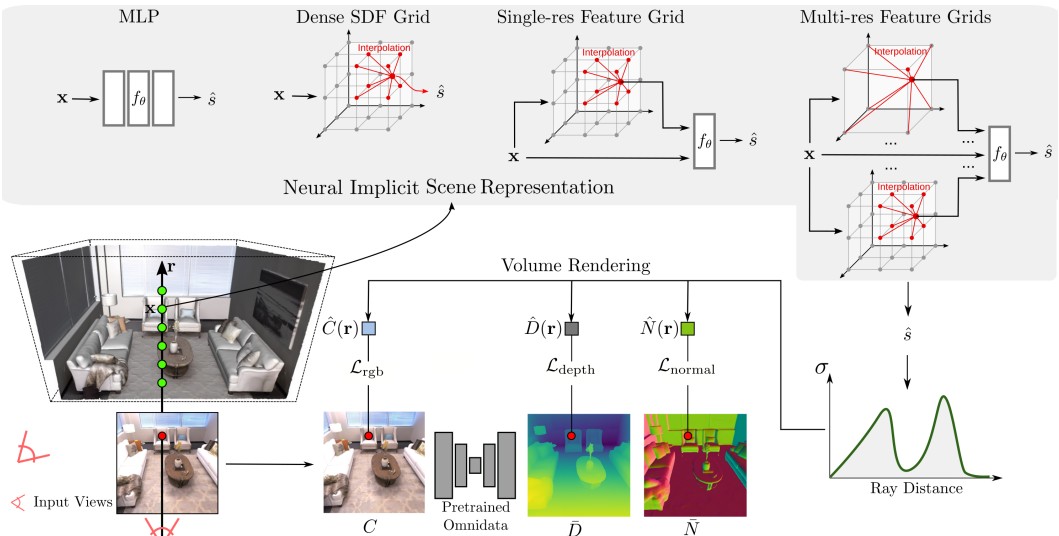

Figure 2: **Overview.** In this work we use monocular geometric cues predicted by a general-purpose pretrained network to guide the optimization of neural implicit surface models. More specifically, for a batch of rays, we volume render predicted RGB colors, depth, and normals, and optimize wrt. the input RGB images and monocular geometric cues. Further, we investigate different design choices for neural implicit architectures and provide an in-depth analysis. For clarity, we only show the SDF and not the color prediction branch above.

## 3 Method

Our goal is to recover the underlying scene geometry from multiple posed images while utilizing monocular geometric cues to guide the optimization process. To this end, we first review neural implicit scene representations and various design choices in Section 3.1 and discuss how to perform volume rendering of these representations in Section 3.2. Next, we introduce the monocular geometric cues we investigate in our study in Section 3.3 and discuss loss functions and the overall optimization process in Section 3.4. An overview of our framework is provided in Fig. 2.

### 3.1 Implicit Scene Representations

We represent scene geometry as a signed distance function (SDF). A signed distance function is a continuous function $f$ that, for a given 3D point, returns the point's distance to the closest surface:

$$f : \mathbb{R}^3 \to \mathbb{R} \qquad \mathbf{x} \mapsto s = \text{SDF}(\mathbf{x}) \ . \tag{1}$$

Here, $\mathbf{x}$ is the 3D point and $s$ denotes the corresponding SDF value. In this work, we parameterize the SDF function with learnable parameters $\theta$ and investigate several different design choices for representing the function: explicit as a dense grid of learnable SDF values, implicit as a single MLP, or hybrid using an MLP in combination with single- or multi-resolution feature grids.

**Dense SDF Grid.** The most straightforward way of parameterizing an SDF is to directly store SDF values in each cell of a discretized volume $\mathcal{G}_\theta$ with resolution of $R_H \times R_W \times R_D$ [28]. To query the SDF value $\hat{s}$ for an arbitrary point $\mathbf{x}$ from the dense SDF grid, we can use any interpolation operation:

$$\hat{s} = \texttt{interp}(\mathbf{x}, \mathcal{G}_\theta) \ . \tag{2}$$

In our experiments, we implement `interp` as trilinear interpolation.

**Single MLP.** The SDF function can also be parameterized by a single MLP [44] $f_\theta$:

$$\hat{s} = f_\theta(\gamma(\mathbf{x})) \ , \tag{3}$$

where $\hat{s}$ is the predicted SDF value and $\gamma$ corresponds to a fixed positional encoding [38,64] mapping $\mathbf{x}$ to a higher dimensional space. After their introduction to novel view synthesis [38], positional

encoding functions are now widely used for neural implicit surface reconstruction [43, 69, 74, 75] as they increase the expressiveness of coordinate-based networks [64].

**Single-Resolution Feature Grid with MLP Decoder.** We can also combine both parameterizations and use a feature-conditioned MLP $f_\theta$ together with a feature grid $\Phi_\theta$ with a resolution of $R^3$, where each cell of the grid stores a feature vector [25, 33, 48, 63] instead of directly storing SDF values:

$$\hat{s} = f_\theta(\gamma(\mathbf{x}), \texttt{interp}(\mathbf{x}, \Phi_\theta)) \ . \tag{4}$$

Note that the MLP $f_\theta$ is conditioned on the interpolated local feature vector from the feature grid $\Phi_\theta$.

**Multi-Resolution Feature Grids with MLP Decoder.** Instead of using a single feature grid $\Phi_\theta$, one can also employ multi-resolution feature grids $\{\Phi_\theta^l\}_{l=1}^L$ with resolutions $R_l$ [9, 19, 39, 63, 82]. The resolutions are sampled in geometric space [39] to combine features at different frequencies:

$$R_l := \lfloor R_{\min} b^l \rfloor \qquad b := \exp\left(\frac{\ln R_{\max} - \ln R_{\min}}{L - 1}\right) \ , \tag{5}$$

where $R_{\min}, R_{\max}$ are the coarsest and finest resolution, respectively. Similarly, we extract the interpolated features at each level and concatenate them together:

$$\hat{s} = f_\theta(\gamma(\mathbf{x}), \{\texttt{interp}(\mathbf{x}, \Phi_\theta^l)\}_l)) \ . \tag{6}$$

As the total number of grid cells grows cubically, we use a fixed number of parameters to store the feature grids and use a spatial hash function to index the feature vector at finer levels [39] (see supplementary for details).

**Color Prediction.** In addition to the 3D geometry, we also predict color values such that our model can be optimized with a reconstruction loss. Following [75], we therefore define a second function $\mathbf{c}_\theta$

$$\hat{\mathbf{c}} = \mathbf{c}_\theta(\mathbf{x}, \mathbf{v}, \hat{\mathbf{n}}, \hat{\mathbf{z}}) \tag{7}$$

that predicts a RGB color value $\hat{\mathbf{c}}$ for a 3D point $\mathbf{x}$ and a viewing direction $\mathbf{v}$. The 3D unit normal $\hat{\mathbf{n}}$ is the analytical gradient of our SDF function. The feature vector $\hat{\mathbf{z}}$ is the output of a second linear head of the SDF network as in [75]. We parameterize $\mathbf{c}_\theta$ with a two-layer MLP with network weights $\theta$. In case of the dense grid SDF parameterization, we similarly optimize a dense feature grid and obtain the feature vector $\hat{\mathbf{z}}$ via the interpolation function $\texttt{interp}$.

## 3.2 Volume Rendering of Implicit Surfaces

Following recent work [43, 69, 74, 75], we optimize the implicit representations described in Section 3.1 via an image-based reconstruction loss using differentiable volume rendering. More specifically, to render a pixel, we cast a ray $\mathbf{r}$ from the camera center $\mathbf{o}$ through the pixel along its view direction $\mathbf{v}$. We sample $M$ points $\mathbf{x}_\mathbf{r}^i = \mathbf{o} + t_\mathbf{r}^i \mathbf{v}$ along the ray and predict their SDF $\hat{s}_\mathbf{r}^i$ and color values $\hat{\mathbf{c}}_\mathbf{r}^i$. We follow [74] to transform the SDF values $\hat{s}_\mathbf{r}^i$ to density values $\sigma_\mathbf{r}^i$ for volume rendering:

$$\sigma_\beta(s) = \begin{cases} \frac{1}{2\beta} \exp\left(\frac{s}{\beta}\right) & \text{if } s \leq 0 \\ \frac{1}{\beta}\left(1 - \frac{1}{2}\exp\left(-\frac{s}{\beta}\right)\right) & \text{if } s > 0 \end{cases} , \tag{8}$$

where $\beta$ is a learnable parameter. Following NeRF [38], the color $\hat{C}(\mathbf{r})$ for the current ray $\mathbf{r}$ is computed via numerical integration:

$$\hat{C}(\mathbf{r}) = \sum_{i=1}^M T_\mathbf{r}^i \alpha_\mathbf{r}^i \hat{\mathbf{c}}_\mathbf{r}^i \qquad T_\mathbf{r}^i = \prod_{j=1}^{i-1}\left(1 - \alpha_\mathbf{r}^j\right) \qquad \alpha_\mathbf{r}^i = 1 - \exp\left(-\sigma_\mathbf{r}^i \delta_\mathbf{r}^i\right) \ , \tag{9}$$

where $T_\mathbf{r}^i$ and $\alpha_\mathbf{r}^i$ denote the transmittance and alpha value of sample point $i$ along ray $\mathbf{r}$, respectively, and $\delta_\mathbf{r}^i$ is the distance between neighboring sample points. Similarly, we compute the depth $\hat{D}(\mathbf{r})$ and normal $\hat{N}(\mathbf{r})$ of the surface intersecting the current ray as:

$$\hat{D}(\mathbf{r}) = \sum_{i=1}^M T_\mathbf{r}^i \alpha_\mathbf{r}^i t_\mathbf{r}^i \qquad \hat{N}(\mathbf{r}) = \sum_{i=1}^M T_\mathbf{r}^i \alpha_\mathbf{r}^i \hat{\mathbf{n}}_\mathbf{r}^i \ . \tag{10}$$

### 3.3 Exploiting Monocular Geometric Cues

Unifying volume rendering with implicit surfaces leads to impressive 3D reconstruction results. Yet, this approach struggles with more complex scenes especially in textureless and sparsely covered regions. To overcome this limitation, we use readily available, efficient-to-compute monocular geometric priors thereby improving neural implicit surface methods.

**Monocular Depth Cues.** One common monocular geometric cue is a monocular depth map, which can be easily obtained via an off-the-shelf monocular depth predictor. More specifically, we use a pretrained Omnidata model [14] to predict a depth map $\bar{D}$ for each input RGB image. Note that the absolute scale is difficult to estimate in general scenes, so $\bar{D}$ must be considered as a relative cue. However, this relative depth information is provided also over larger distances in the image.

**Monocular Normal Cues.** Another geometric cue we use is the surface normal. Similar to the depth cues, we apply the same pretrained Omnidata model to acquire a normal map $\bar{N}$ for each RGB image. Unlike depth cues that provide semi-local relative information, normal cues are local and capture geometric detail. We hence expect that surface normals and depth are complementary to each other.

### 3.4 Optimization

**Reconstruction Loss.** Eq. (9) provides a linkage from the 3D scene representation to 2D observations. We can therefore optimize the scene representation with a simple RGB reconstruction loss:

$$\mathcal{L}_{\text{rgb}} = \sum_{\mathbf{r} \in \mathcal{R}} \|\hat{C}(\mathbf{r}) - C(\mathbf{r})\|_1 \ . \tag{11}$$

Here $\mathcal{R}$ denotes the set of pixels/rays in the minibatch and $C(\mathbf{r})$ is the observed pixel color.

**Eikonal Loss.** Following common practice, we also add an Eikonal term [17] on the sampled points to regularize SDF values in 3D space

$$\mathcal{L}_{\text{eikonal}} = \sum_{\mathbf{x} \in \mathcal{X}} (\|\nabla f_\theta(\mathbf{x})\|_2 - 1)^2 \ , \tag{12}$$

where $\mathcal{X}$ are a set of uniformly sampled points together with near-surface points [74].

**Depth Consistency Loss.** Besides $\mathcal{L}_{\text{rgb}}$ and $\mathcal{L}_{\text{eikonal}}$, we also enforce consistency between our rendered expected depth $\hat{D}$ and the monocular depth $\bar{D}$:

$$\mathcal{L}_{\text{depth}} = \sum_{\mathbf{r} \in \mathcal{R}} \|(w\hat{D}(\mathbf{r}) + q) - \bar{D}(\mathbf{r})\|^2 \ , \tag{13}$$

where $w$ and $q$ are the scale and shift used to align $\hat{D}$ and $\bar{D}$ since $\bar{D}$ is defined only up to scale. Note that these factors have to be estimated individually per batch as the depth maps predicted for different batches can differ in scale and shift. Specifically, we solve for $w$ and $q$ with a least-squares criterion [16, 50] which has a closed-form solution (see supplementary for details).

**Normal Consistency Loss.** Similarly, we impose consistency on the volume-rendered normal $\hat{N}$ and the predicted monocular normals $\bar{N}$ transformed to the same coordinate system with angular and L1 losses [14]:

$$\mathcal{L}_{\text{normal}} = \sum_{\mathbf{r} \in \mathcal{R}} \|\hat{N}(\mathbf{r}) - \bar{N}(\mathbf{r})\|_1 + \|1 - \hat{N}(\mathbf{r})^\top \bar{N}(\mathbf{r})\|_1 \ . \tag{14}$$

The overall loss we use to optimize our implicit surfaces jointly with the appearance network is:

$$\mathcal{L} = \mathcal{L}_{\text{rgb}} + \lambda_1 \mathcal{L}_{\text{eikonal}} + \lambda_2 \mathcal{L}_{\text{depth}} + \lambda_3 \mathcal{L}_{\text{normal}} \ . \tag{15}$$

**Implementation Details.** We implement our method in PyTorch [46] and use the Adam optimizer [29] with a learning rate of 5e-4 for neural networks and 1e-2 for feature grids and dense SDF grids. We set $\lambda_1$, $\lambda_2$, $\lambda_3$ to 0.1, 0.1, 0.05, respectively. We sample 1024 rays per iteration and apply the error-bounded sampling strategy introduced by [74] to sample points along each ray. For MLPs and feature grids, we adapt the architecture and initialization scheme from [74] and [39], respectively. For obtaining monocular cues, we first resize each image and center crop it to $384 \times 384$, which we then feed as input to the pretrained Omnidata model [14]. See supplementary for more details.

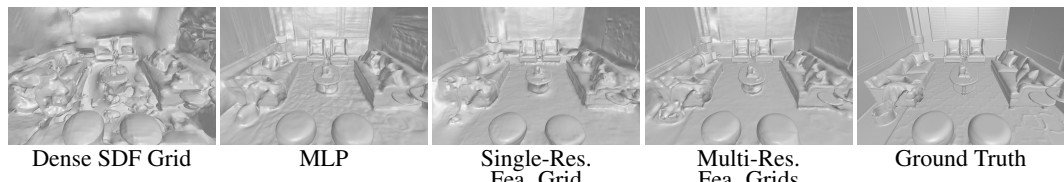

| Dense SDF Grid | MLP | Single-Res.
Fea. Grid | Multi-Res.
Fea. Grids | Ground Truth |

Figure 3: **Architectural Ablation Study.** Comparing different design choices for neural implicit surface representations, we observe that a dense SDF grid leads to noisy reconstructions due to a missing smoothness bias. The MLP and the Single-Res. Fea. Grid improve results, but geometry tends to be overly smooth with missing details. The best results are obtained using Multi-Res. Fea. Grids.

## 4 Experiments

We first analyze different architectural design choices and perform ablation studies wrt. monocular cues and optimization time on a room-level dataset (Replica) with perfect ground truth. Next, we provide qualitative and quantitative comparisons against state-of-the-art baselines on real-world indoor scenes. Finally, we evaluate our method on object-level reconstruction for both sparse input and dense input scenarios.

**Datasets.** While previous neural implicit-based reconstruction methods mainly focused on single-object scenes with many input views, in this work, we investigate the importance of monocular geometric cues for scaling to more complex scenes. Thus we consider: a) Real-world indoor scans: Replica [61] and ScanNet [12]; b) Real-world large-scale indoor scenes: Tanks and Temples [30] advanced scenes; c) Object-level scenes: DTU [1] in the sparse 3-view setting from [40, 76].

**Baselines.** We compare against a) state-of-the-art neural implicit surfaces methods: UNISURF [43], VolSDF [74], NeuS [69], and Manhattan-SDF [18]. b) Classic MVS methods: COLMAP [56] and a state-of-the-art commercial software (RealityCapture[2]). c) TSDF-Fusion [11] with predicted monocular depth cues, where GT depth maps are used to recover the scale and shift values (cf. Eq. (13)). This baseline shows the reconstruction quality if only monocular depth cues and no implicit surface model is used.

**Evaluation Metrics.** For DTU, we follow the official evaluation protocol and report the Chamfer distance. For Replica and ScanNet, following [18, 37, 47, 48, 62, 82], we report the Chamfer Distance, the F-score with a threshold of 5cm, as well as a Normal Consistency measure.

### 4.1 Ablation Study

We first analyze different scene representation choices on the Replica dataset. Next, we ablate the impact of our geometric cues on reconstruction quality and convergence time.

**Architecture Choices for Scene Representations.** We compare the four different scene geometry representations introduced in Section 3.1 and report metrics averaged over the Replica dataset in Table 1. Note that no monocular geometric cues are used here. We first observe that using a single MLP as the scene geometry representation leads to decent results, but the recon-

|  | Normal C.↑ | Chamfer-$L_1$ ↓ | F-score ↑ |
|---|---|---|---|
| MLP [74] | 86.48 | 6.75 | 66.88 |
| Dense SDF Grid | 57.30 | 26.68 | 15.50 |
| Single-res. Fea. Grid | 86.41 | 6.28 | 64.22 |
| Multi-res. Fea. Grids | **87.95** | **5.03** | **78.38** |

Table 1: **Architectural Ablation on Replica.**

struction tends to be over-smooth (see Table 1 and Fig. 3). For grid-based representations, optimizing a dense SDF grid leads to a significantly worse performance compared to all other neural implicit scene representations, even with careful parameter tuning. The reason is the lack of a smoothness bias: The SDF values in grid cells are all stored and optimized independently of each other, hence there is no local or global smoothness bias. In contrast, the Single-Res. Fea. Grid replaces the SDF value in each grid cell with a low-dimensional latent code, and uses a shallow MLP conditioned on these features to read out SDF values of arbitrary 3D points. This modification leads to a notable boost in reconstruction quality over the dense grid, performing similarly well as the single MLP. Using a Multi-Res. Fea. Grids as in [39] further increases performance. We observe that the Multi-Res. Fea. Grids is the best-performing grid-based model, and from now on we report results for the single MLP

---

[2]https://www.capturingreality.com/

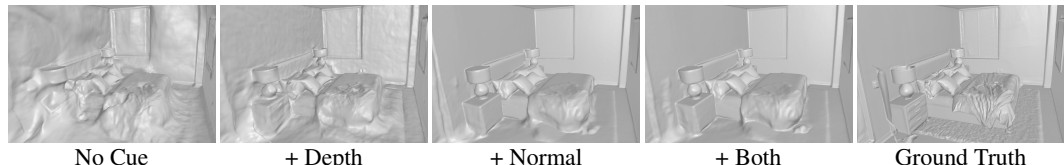

| No Cue | + Depth | + Normal | + Both | Ground Truth |

Figure 4: **Ablation of Monocular Geometric Cues.** Monocular geometric cues significantly improve reconstruction quality for both architectures (we show our MLP variant). With monocular depth cues, the recovered geometry contains more details and a better overall structure. With normal cues, missing details are added and the results become smoother. Using both cues leads to the best performance.

|  |  | Normal C.↑ | Chamfer-$L_1$ ↓ | F-score ↑ |
|---|---|---|---|---|
| **MLP** | No Cues | 86.48 | 6.75 | 66.88 |
|  | Only Depth | 90.56 | 4.26 | 76.42 |
|  | Only Normal | 91.35 | 3.19 | 85.84 |
|  | Both Cues | **92.11** | **2.94** | **86.18** |
| **Multi-Res. Grids** | No Cues | 87.95 | 5.03 | 78.38 |
|  | Only Depth | 90.87 | 3.75 | 80.32 |
|  | Only Normal | 89.90 | 3.61 | 81.28 |
|  | Both Cues | **90.93** | **3.23** | **85.91** |

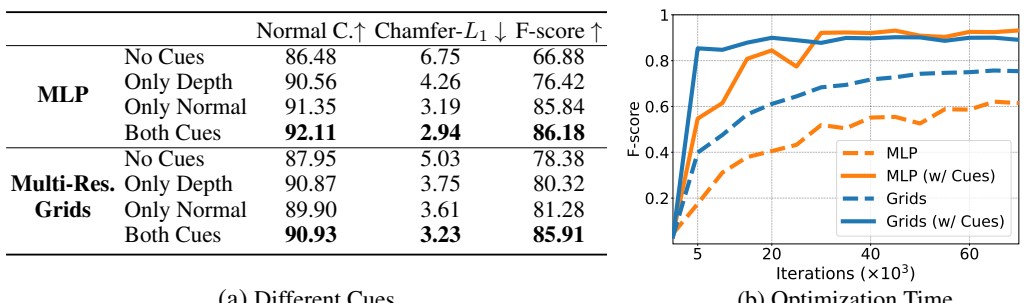

(a) Different Cues          (b) Optimization Time

Table 2: **Ablation of Monocular Geometric Cues.** a.) We report reconstruction results on Replica for MLP and Multi-Res. Grids with and without the monocular geometric cues. We observe that monocular cues improve reconstruction quality for both architectures, and using both cues in combination leads to the best performance. b.) The optimization speed becomes significantly faster when incorporating monocular cues. Comparing the two architectures, we observe that the grid approach yields faster convergences while the MLP with both cues leads to the best results.

and the Multi-Res. Feature Grids. For simplicity, we will refer to the multi-resolution feature grids as *Multi-Res. Grids* or *Grids* in the following.

**Ablation of Different Cues.** We now investigate the effectiveness of different monocular geometric cues for the two chosen representations. Table 2 (a) and Fig. 4 show that, for both representations, using either one or both monocular cues significantly boosts reconstruction quality. We also find both cues to be complementary, with the best performance being achieved when using both. Similar behavior can be observed for the other two representations (cf. supplementary material). It is worth noting that the differences between the two representations become negligible when using monocular cues, indicating that those serve as a general drop-in to improve reconstruction quality.

**Optimization Time.** Table 2 (b) shows optimization time for the two scene representations with and without cues. We see that the Multi-Res. Grids converge faster than the single MLP model. Further, adding the monocular cues significantly speeds up the convergence process. After only 10K iterations, both representations perform better than the converged models without monocular cues. Note that the overhead required for incorporating the monocular cues into the optimization process is small and can be neglected. An extended version of Table 2 (b) can be found in the supplementary materials.

## 4.2 Real-world Large-scale Scene Reconstruction

To show the effectiveness of our method for large-scale scene reconstruction, we compare against various baselines on two challenging large-scale indoor datasets.

**ScanNet.** On ScanNet, we use the test split from [18] and also follow their evaluation protocol in which depth maps are rendered from input camera poses and then re-fused using TSDF Fusion [11] to evaluate only observed areas. We observe in Table 3 that our MLP variant outperforms all baselines achieving smoother reconstructions with more fine details. Note that we outperform concurrent work [68]. Further, we find that the MLP variant performs significantly better than using Multi-Res. Grids. ScanNet's RGB images contain motion blur and the camera poses are also noisy. This can be harmful to the local geometry updates in grid-based representations, while MLPs are more robust to this noise due to their smoothness bias.

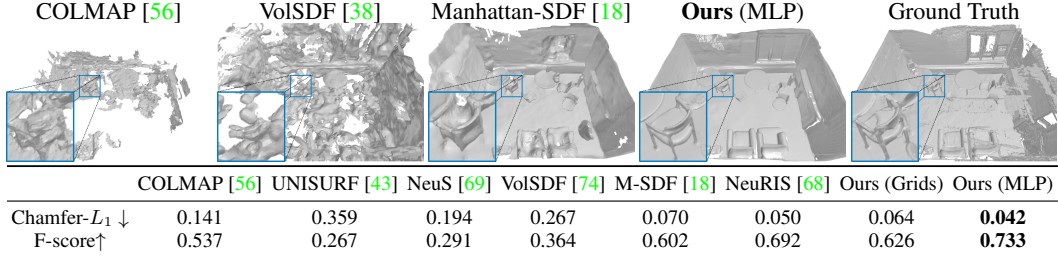

| | COLMAP [56] | UNISURF [43] | NeuS [69] | VolSDF [74] | M-SDF [18] | NeuRIS [68] | Ours (Grids) | Ours (MLP) |
|---|---|---|---|---|---|---|---|---|
| Chamfer-$L_1$ ↓ | 0.141 | 0.359 | 0.194 | 0.267 | 0.070 | 0.050 | 0.064 | **0.042** |
| F-score↑ | 0.537 | 0.267 | 0.291 | 0.364 | 0.602 | 0.692 | 0.626 | **0.733** |

Table 3: **Scene-level Reconstruction on ScanNet.** Colmap and VolSDF do not lead to competitive reconstructions. Manhatten-SDF achieves compelling results, but less-observed areas are noisier and details are missing. In contrast, our approaches reconstruct smooth and details surfaces, achieving the best results. Further, MLPs are more robust to the motion blur and noise in camera poses.

**Tanks & Temples.** To further investigate the scalability of our method to larger-scale scenes, we conduct experiments on the Tanks and Temples advanced sets. The qualitative results in Fig. 1 show that the monocular cues significantly boost the performance of VolSDF [74], making MonoSDF the first neural implicit model achieving reasonable results on such a large-scale indoor scene. See the supplementary material for more visual comparisons and discussions.

### 4.3  Object-level Reconstruction from Sparse Views

We now evaluate our method on another challenging task: reconstructing single objects from sparse input views. We adopt the test split from [74,75] on DTU and choose *three* input views following [40].

We first observe in Table 4 and Fig. 1 that without the usage of the monocular geometric cues, neither the MLP (VolSDF [74]) nor the Multi-Res. Grids work well with only 3 input views. When incorporating the cues, the results for both representations are significantly improved. Interestingly, the grid-based representations perform inferior to a single MLP as they are updated locally and do not benefit from the inductive bias of a monolithic MLP representation.

Comparing against TSDF Fusion [11] that fused predicted depth cues from all views into a TSDF volume without any optimization, we observe that this baseline has difficulties in reconstructing meaningful details due to inconsistencies in the monocular depth cues. Note that this baseline uses the GT depth maps from [13] to compute scale and shift for the depth cues. Classic MVS methods

| | Chamfer-$L_1$ ↓ |
|---|---|
| TSDF-Fusion [11] | 4.80 |
| COLMAP [56] | 2.56 |
| RealityCapture | 2.84 |
| Grids | 6.47 |
| Grids w/ cues | 3.68 |
| MLP [74] | 4.21 |
| MLP w/ cues | **1.86** |

Table 4: **Reconstruction on DTU (3 Views).** We report the average over the test split from [74] (see supplementary for per-object results).

perform well quantitatively, but they heavily rely on dense matching, and in case of three input images, this inevitably leads to incomplete reconstructions (see supplementary material). In contrast, our approach combines neural implicit surface representations with the benefits from monocular geometric cues that are more robust to less-observed regions.

### 4.4  Object-level Reconstruction from Dense Views

To further investigate the effectiveness and flexibility of our method, we evaluate our approach on the DTU dataset with all input views, which is a common setting in recent work [43,70,74]. In this experiment, we simply resize the low-resolution monocular cues to full resolution (from $384 \times 384$ to $1200 \times 1200$ pixels) while keeping the image ratio. As the original image is of size $1200 \times 1600$, the monocular cues are missing in the left and right part of the image. Therefore, we only use the monocular cues where they are available.

As shown in Table 5, our approach with MLP architecture achieves reconstruction quality similar to state-of-the-art methods [43,70,74]. This is reasonable as the dense input views provide enough constraints and the prior information from monocular cues is negligible. However, our method with multi-resolution feature grid architecture outperforms previous work by a large margin. We attribute this to the expressiveness of multi-resolution feature grids where monocular cues are still effective to

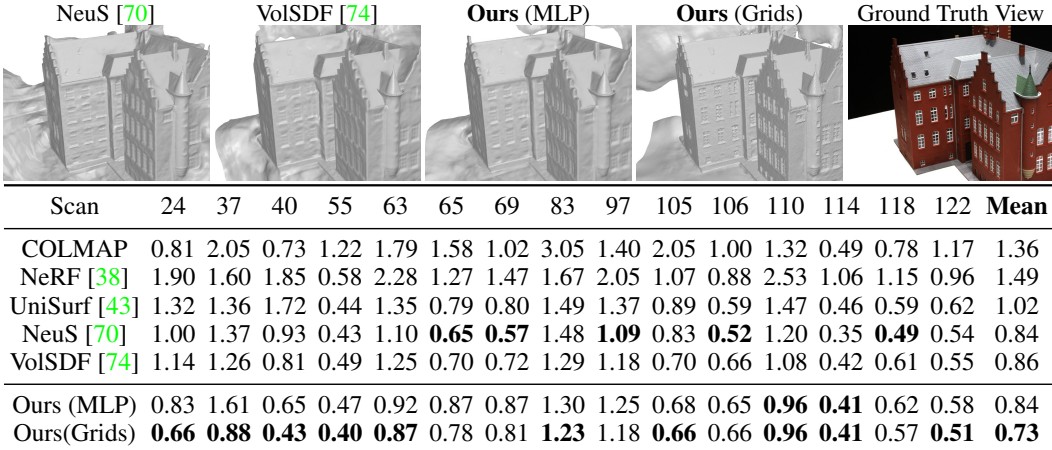

| Scan | 24 | 37 | 40 | 55 | 63 | 65 | 69 | 83 | 97 | 105 | 106 | 110 | 114 | 118 | 122 | **Mean** |
|---|---|---|---|---|---|---|---|---|---|---|---|---|---|---|---|---|
| COLMAP | 0.81 | 2.05 | 0.73 | 1.22 | 1.79 | 1.58 | 1.02 | 3.05 | 1.40 | 2.05 | 1.00 | 1.32 | 0.49 | 0.78 | 1.17 | 1.36 |
| NeRF [38] | 1.90 | 1.60 | 1.85 | 0.58 | 2.28 | 1.27 | 1.47 | 1.67 | 2.05 | 1.07 | 0.88 | 2.53 | 1.06 | 1.15 | 0.96 | 1.49 |
| UniSurf [43] | 1.32 | 1.36 | 1.72 | 0.44 | 1.35 | 0.79 | 0.80 | 1.49 | 1.37 | 0.89 | 0.59 | 1.47 | 0.46 | 0.59 | 0.62 | 1.02 |
| NeuS [70] | 1.00 | 1.37 | 0.93 | 0.43 | 1.10 | **0.65** | **0.57** | 1.48 | **1.09** | 0.83 | **0.52** | 1.20 | 0.35 | **0.49** | 0.54 | 0.84 |
| VolSDF [74] | 1.14 | 1.26 | 0.81 | 0.49 | 1.25 | 0.70 | 0.72 | 1.29 | 1.18 | 0.70 | 0.66 | 1.08 | 0.42 | 0.61 | 0.55 | 0.86 |
| Ours (MLP) | 0.83 | 1.61 | 0.65 | 0.47 | 0.92 | 0.87 | 0.87 | 1.30 | 1.25 | 0.68 | 0.65 | **0.96** | **0.41** | 0.62 | 0.58 | 0.84 |
| Ours(Grids) | **0.66** | **0.88** | **0.43** | **0.40** | **0.87** | 0.78 | 0.81 | **1.23** | 1.18 | **0.66** | 0.66 | **0.96** | **0.41** | 0.57 | **0.51** | **0.73** |

Table 5: **Object-level Reconstruction on DTU Dataset will All Input Views.** We compare Chamfer distance with state-of-the-art methods. Our approach with MLP achieves similar results to previous methods, while our method with multi-resolution feature grids leads to more detailed surfaces and outperforms previous work by a large margin.

suppress noise and therefore can reconstruct smooth and detailed surfaces. We kindly refer the reader to the supplementary material for additional visual comparisons.

## 5  Conclusion

We have presented MonoSDF, a novel framework that systematically explores how monocular geometric cues can be incorporated into the optimization of neural implicit surfaces from multi-view images. We show that such easy-to-obtain monocular cues can significantly improve 3D reconstruction quality, efficiency, and scalability for a variety of neural implicit representations. When using monocular cues, a simple MLP architecture performs best overall, demonstrating that MLPs in principle are able to represent complex scenes, albeit being slower to converge compared to grid-based representations. Multi-resolution feature grids in general can converge fast and capture details, but are less robust to noise and ambiguities in the input images.

**Limitations.**  The performance of our model depends on the quality of the monocular cues. Filtering strategies to handle failures of the monocular predictor are thus a promising direction to further improve reconstruction quality. We kindly refer the reader to the supplementary material for additional analysis. While we demonstrated that integrating depth and normal cues significantly improves reconstruction, exploring other cues such as occlusion edges, plane, or curvature [14, 78] is an interesting future direction. We are currently limited by the low-resolution ($384 \times 384$ pixels) output of the Omnidata model [14] and plan to explore different ways of using higher-resolution cues. We provide some preliminary results of using high-resolution cues in the supplementary. Joint optimization of scene representations and camera parameters [3, 82] is another interesting direction, especially for multi-resolution grids, in order to better handle noisy camera poses.

## Acknowledgments and Disclosure of Funding

This work was supported by an NVIDIA research gift. We thank the Max Planck ETH Center for Learning Systems (CLS) for supporting SP and the International Max Planck Research School for Intelligent Systems (IMPRS-IS) for supporting MN. ZY is supported by BMWi in the project KI Delta Learning (project number 19A19013O). AG is supported by the ERC Starting Grant LEGO-3D (850533) and DFG EXC number 2064/1 - project number 390727645. TS is supported by the EU Horizon 2020 project RICAIP (grant agreeement No.857306), and the European Regional Development Fund under project IMPACT (No. CZ.02.1.01/0.0/0.0/15_003/0000468). We thank the authors of Manhattan-SDF and NeuRIS for sharing results on ScanNet. We also thank Christian Reiser and Zijian Dong for proofreading.

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
