# Supplementary Material for
# MonoSDF: Exploring Monocular Geometric Cues
# for Neural Implicit Surface Reconstruction

**Zehao Yu**[1]  **Songyou Peng**[2,3]  **Michael Niemeyer**[1,3]  **Torsten Sattler**[4]  **Andreas Geiger**[1,3]

[1]University of Tübingen    [2]ETH Zurich    [3]MPI for Intelligent Systems, Tübingen
[4]Czech Technical University in Prague

https://niujinshuchong.github.io/monosdf

In this **supplementary document**, we first discuss architectural and implementation details in Section 1. Next, we provide additional ablation studies of our monocular geometric cues for four different scene representations in Section 2 and report additional quantitative and qualitative results in Section 3. Finally, we discuss potential negative impact of this work in Section 4.

## 1 Implementation Details

In this section, we first present an overview of 4 different architectures for neural implicit scene representations and details of Multi-Res. Grids in Section 1.1 and provide details of the depth loss computation in Section 1.2. Next, we describe additional details regarding our parameterizations and optimization in Section 1.3 and discuss evaluation metrics in Section 1.4.

### 1.1 Architectures

In the main paper, we investigate four different architectures as our scene representation: *Dense SDF Grid*, *Single MLP*, *Single-Res. Grid*, and *Multi-Res. Grids* . See Fig. 1 for an overview over the architectures. In the following, we provide details for Multi-Res. Feature Grids.

**Multi-Res. Grids.** Following Instant-NGP [16], we use $L$ levels of feature grids with resolutions sampled in geometric space to combine features at different frequencies:

$$R_l := \lfloor R_{\min} b^l \rfloor \qquad b := \exp\left(\frac{\ln R_{\max} - \ln R_{\min}}{L - 1}\right) \ , \tag{1}$$

where $R_{\min}, R_{\max}$ are the coarsest and finest resolutions, respectively. As the total number of grid cells grows cubically, we use a fixed number of parameters to store the feature grids and use a spatial hash function to index the feature vector at finer levels. More specifically, each grid contains up to $T$ feature vectors with dimensionality $F$. At the coarse level where $R_l^3 \leq T$, the feature grid is stored densely. At the finer level where $R_l^3 > T$, a spatial hash function [24] is used to index the corresponding feature vector:

$$h(\mathbf{x}) = \left(\bigoplus_{i=1}^{3} \mathbf{x}_i \pi_i\right) \mod T \ , \tag{2}$$

where $\bigoplus$ is the bit-wise XOR operation and $\pi_i$ are unique, large prime numbers. We use the default values $R_{\min} = 16$, $R_{\max} = 2048$, $L = 16$, $F = 2$, and $T = 2^{19}$ similar to [16] in all experiments.

36th Conference on Neural Information Processing Systems (NeurIPS 2022).

## 1.2 Depth Consistency Loss

We enforce consistency between our rendered expected depth $\hat{D}$ and the monocular depth $\bar{D}$ with a scale invariant loss function:

$$\mathcal{L}_{\text{depth}} = \sum_{\mathbf{r} \in \mathcal{R}} \left\| (w\hat{D}(\mathbf{r}) + q) - \bar{D}(\mathbf{r}) \right\|^2 , \tag{3}$$

where $w$ and $q$ are the scale and shift used to align $\hat{D}$ and $\bar{D}$ since $\bar{D}$ is given only up to scale. Specifically, we solve $w$ and $q$ with a least-squares criterion [8, 20]:

$$(w, q) = \arg\min_{w,q} \sum_{\mathbf{r} \in \mathcal{R}} \left( w\hat{D}(\mathbf{r}) + q - \bar{D}(\mathbf{r}) \right)^2 . \tag{4}$$

$w$ and $q$ can be efficiently computed as follows: Let $\mathbf{h} = (w, q)^T$ and $\mathbf{d_r} = (\hat{D}(\mathbf{r}), 1)^T$, then Eq. (4) can be rewrite as:

$$\mathbf{h}^{\text{opt}} = \arg\min_{\mathbf{h}} \sum_{\mathbf{r} \in \mathcal{R}} \left( \mathbf{d_r}^T \mathbf{h} - \bar{D}(\mathbf{r}) \right)^2 . \tag{5}$$

which has the closed-form solution:

$$\mathbf{h} = \left( \sum_{\mathbf{r}} \mathbf{d_r} \mathbf{d_r}^T \right)^{-1} \left( \sum_{\mathbf{r}} \mathbf{d_r} \bar{D}(\mathbf{r}) \right) . \tag{6}$$

Note that we estimate $w$ and $q$ individually at each iteration for a batch of randomly sampled rays within a single image because depth maps predicted by the monocular depth predictor can differ in scale and shift and the underlying scene geometry changes at each iteration.

## 1.3 Additional Details

For our single MLP architecture, we use an 8-layer MLP with hidden dimension 256. We use a two-layer MLP with hidden dimension 256 for the SDF prediction for both, Single-Res. Grid and Multi-Res. Grids. We implement the color network with a two-layer MLP with hidden dimension 256 and use it for all architectures. We use Softplus activation for geometric network and use ReLU activation for the color network. We explicitly initialize the SDF grid with a sphere and use the geometric initialization from [2] for other architectures. For obtaining monocular cues, we first resize each image and center crop it to $384 \times 384$, which we then feed as input to the pretrained Omnidata model [7]. The output depth and normal maps have the same resolution of $384 \times 384$. As a result, we use the same resolution for RGB images, depth cues and normal cues and adjust camera intrinsics accordingly for all experiments. We optimize our model for 200k iterations which takes about 6 hours and 11 hours for our Multi-Res. Grids and MLP, respectively, on a single NVIDIA RTX3090 GPU.

## 1.4 Evaluation Metrics

For the DTU dataset [1], we follow the official evaluation protocol and report the reconstruction quality with: *Accuracy*, *Completeness* and *Chamfer Distance*. *Accuracy* measures how close the reconstructed points are to the ground truth and is defined as the mean distance of the reconstructed points to the ground truth. *Completeness* measures to what extent the ground truth points are recovered and is defined as the mean distance of the ground truth points to the reconstructed points. *Chamfer Distance* is the mean of *Accuracy* and *Completeness*. It measures the overall reconstruction quality. For efficiency, we use the Python script[1] to compute these evaluation metrics.

For Replica [22] and ScanNet [4], we report *Accuracy*, *Completeness*, *Chamfer Distance*, *Precision*, *Recall*, and *F-score* with a threshold of 5cm following [9, 23, 32]. We further report *Normal Consistency* for the Replica dataset following [9, 13, 18, 19, 23, 32] as near-perfect ground truth is available. These metrics are defined in Table 1.

For the Tanks and Temples dataset [11], we submit our reconstruction results to the official evaluation server[2] and report the provided F-score.

---

[1] https://github.com/jzhangbs/DTUeval-python
[2] https://www.tanksandtemples.org/

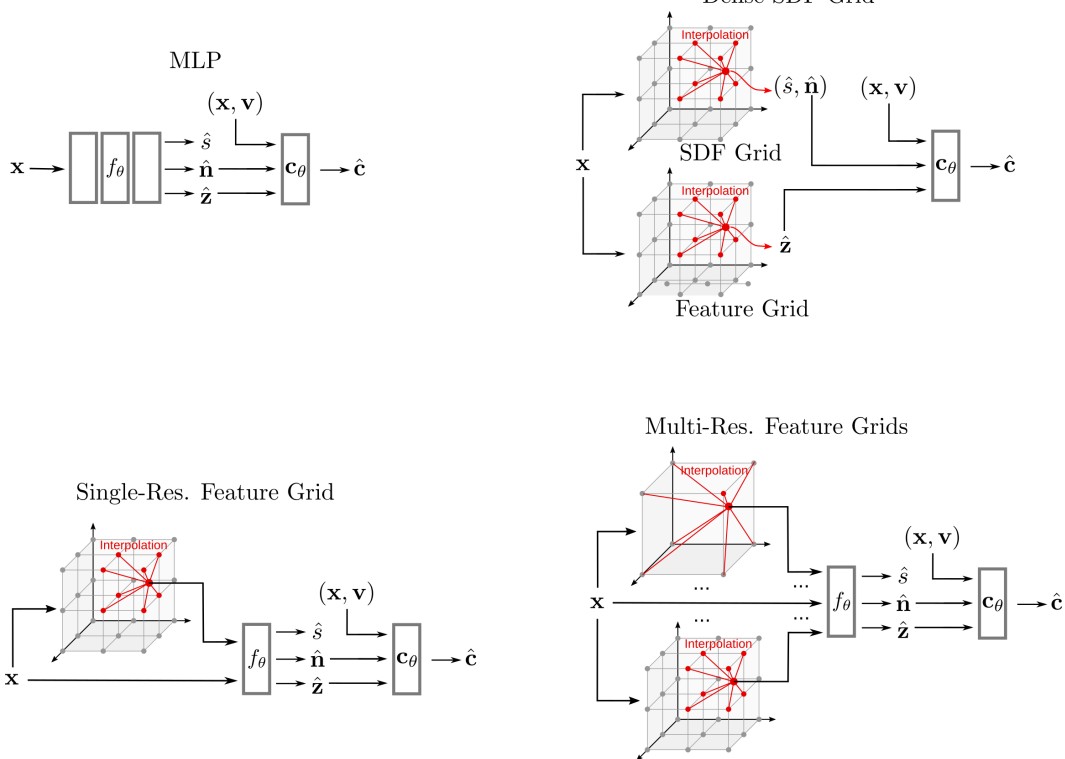

Figure 1: **Architectures.** We show an overview over four different scene representations considered in this paper.

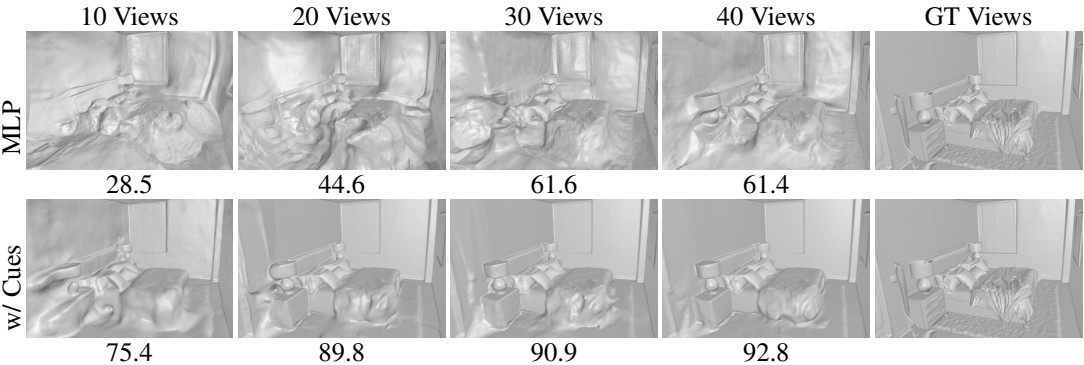

Figure 2: **Ablation of Different Number of Input Views on the Replica Dataset.** We show F-score under each image. We observe that using more input views for training improves reconstruction quality. Further, adding monocular geometric cues improves reconstruction quality. When using only 10 input views, the MLP fails to reconstruct reasonable results while using monocular geometric cues significantly improves results.

## 2 Ablation

In this section, we first conduct several ablation studies to verify the effectiveness of our method, including using geometric cues with different scene representations in Section 2.1, different architecture configurations in Section 2.2, different number of input views in Section 2.3, different cues predictors in Section 2.4. Next, we analyze the optimization time of our framework in Section 2.5.

| Metric | Definition |
|---|---|
| Acc | $\displaystyle \operatorname*{mean}_{\mathbf{p}\in P}\left(\min_{\mathbf{p}^{*}\in P^{*}}||\mathbf{p}-\mathbf{p}^{*}||_{1}\right)$ |
| Comp | $\displaystyle \operatorname*{mean}_{\mathbf{p}^{*}\in P^{*}}\left(\min_{\mathbf{p}\in P}||\mathbf{p}-\mathbf{p}^{*}||_{1}\right)$ |
| Chamfer | $\frac{\text{Acc+Comp}}{2}$ |
| Precision | $\displaystyle \operatorname*{mean}_{\mathbf{p}\in P}\left(\min_{\mathbf{p}^{*}\in P^{*}}||\mathbf{p}-\mathbf{p}^{*}||_{1}<0.05\right)$ |
| Recall | $\displaystyle \operatorname*{mean}_{\mathbf{p}^{*}\in P^{*}}\left(\min_{\mathbf{p}\in P}||\mathbf{p}-\mathbf{p}^{*}||_{1}<0.05\right)$ |
| F-score | $\frac{2\cdot\text{Precision}\cdot\text{Recall}}{\text{Precision+Recall}}$ |
| Normal-Acc | $\displaystyle \operatorname*{mean}_{\mathbf{p}\in P}\left(\mathbf{n}_{\mathbf{p}}^{T}\mathbf{n}_{\mathbf{p}^{*}}\right)\ \text{s.t.}\ \mathbf{p}^{*}=\operatorname*{argmin}_{p^{*}\in P^{*}}||\mathbf{p}-\mathbf{p}^{*}||_{1}$ |
| Normal-Comp | $\displaystyle \operatorname*{mean}_{\mathbf{p}^{*}\in P^{*}}\left(\mathbf{n}_{\mathbf{p}}^{T}\mathbf{n}_{\mathbf{p}^{*}}\right)\ \text{s.t.}\ \mathbf{p}=\operatorname*{argmin}_{p\in P}||\mathbf{p}-\mathbf{p}^{*}||_{1}$ |
| Normal-Consistency | $\frac{\text{Normal-Acc+Normal-Comp}}{2}$ |

Table 1: **Evaluation Metrics.** We show the evaluation metrics with their definitions that we use to measure reconstruction quality. $P$ and $P^{*}$ are the point clouds sampled from the predicted and the ground truth mesh. $\mathbf{n}_{\mathbf{p}}$ is the normal vector at point $\mathbf{p}$.

| | | Test Split | | | Train Split | | |
|---|---|---|---|---|---|---|---|
| | | Normal C.↑ | Chamfer-$L_1$ ↓ | F-score ↑ | Normal C.↑ | Chamfer-$L_1$ ↓ | F-score ↑ |
| **Dense SDF Grid** | No Cues | 57.30 | 26.68 | 15.50 | 60.86 | 17.34 | 26.34 |
| | Only Depth | 71.81 | 12.60 | 30.09 | 73.15 | 13.09 | 30.30 |
| | Only Normal | 73.95 | 13.62 | 33.34 | 77.80 | 11.30 | **42.45** |
| | Both Cues | **76.47** | **11.39** | **37.27** | **80.05** | **10.09** | 41.57 |
| **MLP** | No Cues | 86.48 | 6.75 | 66.88 | 86.69 | 7.48 | 63.24 |
| | Only Depth | 90.56 | 4.26 | 76.42 | 91.80 | 3.59 | 85.67 |
| | Only Normal | 91.35 | 3.19 | 85.84 | 92.85 | 4.23 | 85.58 |
| | Both Cues | **92.11** | **2.94** | **86.18** | **93.86** | **2.63** | **92.12** |
| **Single-Res. Grids** | No Cues | 86.41 | 6.28 | 64.22 | 86.54 | 6.63 | 67.26 |
| | Only Depth | 90.50 | 3.94 | 78.42 | 91.3 | 3.29 | 86.34 |
| | Only Normal | 89.60 | 4.07 | 76.47 | **91.87** | 3.13 | 85.96 |
| | Both Cues | **90.59** | **3.56** | **83.34** | **91.87** | **2.98** | **88.23** |
| **Multi-Res. Grids** | No Cues | 87.95 | 5.03 | 78.38 | 87.15 | 5.83 | 72.13 |
| | Only Depth | 90.87 | 3.75 | 80.32 | 91.25 | 3.41 | **87.04** |
| | Only Normal | 89.90 | 3.61 | 81.28 | 91.11 | 3.59 | 84.02 |
| | Both Cues | **90.93** | **3.23** | **85.91** | **91.41** | **3.14** | 86.87 |

Table 2: **Ablation of Monocular Geometric Cues on Replica.** Our monocular geometric cues significantly improve reconstruction quality across all architectures.

## 2.1 Ablation of Different Cues

To evaluate the effectiveness of our monocular geometric cues for different scene representations, we conduct ablation studies on the Replica dataset with our four different scene representations. Note that as the Replica dataset is part of the training set of Omnidata (making up 0.46% of the entire training data) [7], we split the evaluation into the train/test split of Omnidata [7].

As shown in Table **??** and Fig. 4, our geometric cues improve reconstruction quality significantly independent of the underlying scene representations. We observe that using both, depth cues and

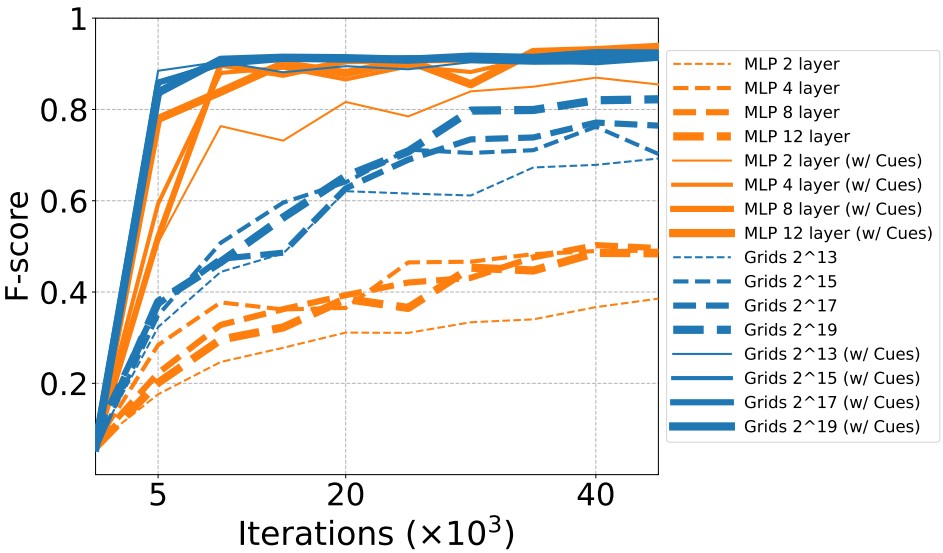

Figure 3: **Optimization Processes Using Different Architecture Configurations.** Using monocular geometric cues improves reconstruction quality and convergence speed independent of the network configurations.

| Model configuration | Num. Params |
|---|---|
| MLP (2 layers) | 0.15M |
| MLP (4 layers) | 0.26M |
| MLP (8 layers) | 0.53M |
| MLP (12 layers) | 0.8M |
| Multi-res. Feature Grids (hash table size $2^{13}$) | 0.41M |
| Multi-res. Feature Grids (hash table size $2^{15}$) | 1.11M |
| Multi-res. Feature Grids (hash table size $2^{17}$) | 3.67M |
| Multi-res. Feature Grids (hash table size $2^{19}$) | 12.67M |

Table 3: **Number of Learnable Parameters Using Different Architecture Configurations.**

normal cues, leads to the best results, indicating the complementary nature of the different cues. We further observe that the reconstruction quality as well as the improvements from adding geometric cues are similar for the train and test split of Omnidata, showing that the monocular predictor did not overfit to the training data.

## 2.2 Ablation of Different Architecture Configurations

In order to evaluate the performance with different model capacities, we consider MLPs with a different number of layers and Multi-res. Feature Grids with different sizes of the hash table. We list the number of learnable parameters using different architecture configurations in the Table 3, and show their performance over the optimization processes in Fig. 3. Our experiments show that using monocular geometric cues improves reconstruction quality and convergence speed independent of the network configuration.

## 2.3 Ablation of Different Numbers of Input Views

We ran experiments with a different number of input images and monocular geometric cues. As shown in Fig. 2, adding the monocular geometric cues leads to consistent improvements across different numbers of input views.

| Method | F-score |
|---|---|
| MLP | 64.2 |
| w/ MiDaS [20] | 68.6 |
| w/ LeReS [29] | 72.6 |
| w/ Omnidata [7] | 86.7 |

(a) Different Depth

| Method | F-score |
|---|---|
| MLP | 64.2 |
| w/ Tilted [6] | 45.6 |
| w/ Omnidata [7] | 92.2 |

(b) Different Normal

| Method | F-score |
|---|---|
| MLP | 64.2 |
| w/ Self-supervised [12,30] | 45.6 |
| w/ Omnidata [7] | 86.7 |

(c) Self-supervised Depth

Table 4: **Ablation of Different Monocular Cues Predictors.** a.) Adding monocular depth improves performance over a single MLP without cues. Unsurprisingly, better depth predictors lead to better performance, with the state-of-the-art Omnidata model giving the best results. b.) Adding monocular normal improve the results. Similarly, using normals predicted by the state-of-the-art Omnidata model leads to the best performance. c.) Using self-supervised depth estimator degrades performance. We hypothesize that this is due to the weaker performance of the self-supervised model which is also trained with an RGB loss and hence suffers from the under-constrained problem of recovering geometry from multi-view images.

## 2.4 Ablation of Different Monocular Cues Predictors

To further analyze the robustness of our approach to monocular geometric cues of different levels of quality, we further tested our model with different supervised depth predictors [20, 29], normal predictors [6], and self-supervised depth predictors [12, 30]. The result is shown in Table 4. We found that using the state-of-the-art Omnidata model leads to the best results, indicating that the development of better geometric cues will further improve the performance of our approach.

## 2.5 Optimization Time

Adding monocular geometric cues to the optimization introduces a small overhead to our overall optimization pipeline. First, predicting these cues with a pretrained Omnidata model is very efficient (36 FPS with an NVIDIA RTX3090 GPU). For example, it takes less than 26 seconds to predict both depth maps and normal maps for 464 images for one of the ScanNet scene. Note that this only needs to be done once and that we measure FPS with a batch size of one; using a larger batch size will result in a speed up. Second, we volume render depth and normals during optimization in order to apply a loss against these monocular cues. This overhead is also small and can be neglected since the most expensive part wrt. compute is the inference of the network. For our MLP variant, the additional flops for volume rendering depth and normal is only 0.0002% of the MLP inference time. While adding monocular geometric cues introduce a small overhead, the improvements in terms of reconstruction quality and converge speed are significant. As shown in Table 2 (b) in the main paper, with only 5k iterations, our Multi-Res. Grids representation with cues performs better than the converged models without geometric cues, which implies a $40\times$ speed up (5k vs. 200k).

# 3 Additional Results

In this section, we provide more qualitative and quantitative results for three datasets: ScanNet ( Section 3.1), Tanks and Temples ( Section 3.2), and DTU ( Section 3.4).

## 3.1 ScanNet

We report quantitative results with all metrics for ScanNet in Table 5 and show more visualizations in Fig. 5. Compared to state-of-the-art methods, our approach with MLP architecture produces significantly better reconstructions both visually as well as quantitatively. It's worth noting that we perform better than concurrent work [25] even though they have some filtering mechanism.

## 3.2 Tanks and Temples

We show quantitative results for Tanks and Temples in Table 6. Qualitative comparisons of with or without monocular cues of our MLP variant are shown in Fig. 6 and Fig. 7. Fig. 8 and Fig. 9 show

|  | Acc↓ | Comp↓ | Chamfer-$L_1$ ↓ | Prec↑ | Recall↑ | F-score↑ |
|---|---|---|---|---|---|---|
| COLMAP [21] | 0.047 | 0.235 | 0.141 | 0.711 | 0.441 | 0.537 |
| UNISURF [17] | 0.554 | 0.164 | 0.359 | 0.212 | 0.362 | 0.267 |
| NeuS [26] | 0.179 | 0.208 | 0.194 | 0.313 | 0.275 | 0.291 |
| VolSDF [28] | 0.414 | 0.120 | 0.267 | 0.321 | 0.394 | 0.346 |
| Manhattan-SDF [9] | 0.072 | 0.068 | 0.070 | 0.621 | 0.586 | 0.602 |
| NeuRIS [25] | 0.050 | 0.049 | 0.050 | 0.717 | 0.669 | 0.692 |
| **Ours** (Multi-Res. Grids) | 0.072 | 0.057 | 0.064 | 0.660 | 0.601 | 0.626 |
| **Ours** (MLP) | **0.035** | **0.048** | **0.042** | **0.799** | **0.681** | **0.733** |

Table 5: **Scene-level 3D Reconstruction on ScanNet.** We report reconstruction results for our methods and baselines on ScanNet (baselines from [9]). We find that our approaches outperform previous state-of-the-art, highlighting the effectiveness of the use of monocular geometric priors. As ScanNet's RGB images contain motion blur and the camera poses are partially noisy, we further observe that the MLP architecture is more robust to this noise and achieves the best results. It's worth noting that we perform better than concurrent work [25] even though they have some filtering mechanism.

|  | Grid | Grid w/ cues | MLP [28] | MLP w/ cues |
|---|---|---|---|---|
| Auditorium | 1.36 | **3.17** | 1.60 | 3.09 |
| Ballroom | 2.67 | **3.70** | 2.04 | 2.47 |
| Courtroom | 7.84 | **13.75** | 8.03 | 10.00 |
| Museum | 4.12 | **5.68** | 2.96 | 5.10 |
| mean | 4.00 | **6.58** | 3.66 | 5.165 |

Table 6: **Evaluation Results on the Tanks and Temples Dataset Advanced Set.** We evaluate the reconstructed meshes using the official server and report the F-score with 10mm. Our monocular geometric cues improve the reconstruction quality for all scenes.

qualitative comparison of our Mulit-Res. Grids. Our monocular geometric cues significantly improve the reconstruction quality.

We further show an additional comparison against state-of-the-art MVS methods in Fig. 10. We use a pretrained Vis-MVSNet [31] to predict depth maps for the input images and fuse them to point clouds follow the official code.[3] Next, we use Meshlab's screened Poisson reconstruction [10] to reconstruct a mesh from point clouds with default parameters. We observe that our reconstructions are more complete which is useful for many applications. Further, reconstructing a mesh from point clouds involves lossy post-processing, leading to floating artifacts and bloated areas in less-observed areas.

### 3.3 Preliminary Results of Using High-resolution Monocular Cues

In the main paper, we center-crop each image and resize it to $384 \times 384$. Then, we use a pretrained Omnidata model to predict depth maps and normal maps which are also of size $384 \times 384$. While we have shown that training at a resolution of $384 \times 384$ produces impressive results, we believe that exploring different ways to generate and integrate higher resolution cues could further improve reconstruction quality. Here, we provide a proof-of-concept experiment for generating higher resolution monocular cues and integrating them into our model. We use a divide-and-conquer method for generating high-resolution cues. First, we partition a high-resolution image to multiple overlapping sub-images, and we predict monocular depth and normal for each sub-image. Next, we merge these predictions. We use Eq. 6 to align the depth maps and solve the rotation for the normal maps. An example of the resulting high-resolution monocular cues is shown in Fig. 11. We found that our high-resolution cues contain more fine details compared to low-resolution cues. Note that using other methods for generating high-resolution depth maps is also possible, e.g., [14]. We then use the high-resolution cues to train our model, and the results are shown in Fig. 12. We observe significant improvements when using high-resolution monocular cues.

---

[3]Available at https://github.com/jzhangbs/Vis-MVSNet

| | TSDF [3] | COLMAP | RealityCapture | MLP [28] | MLP w/ cues | Multi-Res. Grids | Multi-Res. Grids w/ cues |
|---|---|---|---|---|---|---|---|
| scan24 | 5.01 | 4.45 | 4.19 | 5.24 | **3.47** | 6.46 | 5.24 |
| scan37 | 5.28 | 4.67 | 3.85 | 5.09 | **3.61** | 8.30 | 6.37 |
| scan40 | 5.09 | 2.51 | 2.26 | 3.99 | **2.10** | 7.03 | 2.52 |
| scan55 | 4.63 | 1.90 | 2.49 | 1.42 | **1.05** | 5.87 | 1.95 |
| scan63 | 5.03 | 2.81 | 3.49 | 5.10 | **2.37** | 6.92 | 6.64 |
| scan65 | 4.50 | 2.92 | 3.97 | 4.33 | **1.38** | 3.09 | 2.05 |
| scan69 | 4.55 | 2.12 | 1.91 | 5.36 | **1.41** | 5.34 | 4.25 |
| scan83 | 4.88 | 2.05 | 2.49 | 3.15 | **1.85** | 6.03 | 1.81 |
| scan97 | 6.22 | 2.93 | 2.37 | 5.78 | **1.74** | 6.93 | 5.27 |
| scan105 | 3.89 | 2.05 | 2.27 | 2.07 | **1.10** | 6.01 | 2.54 |
| scan106 | 5.67 | 2.01 | 2.90 | 2.79 | **1.46** | 6.14 | 3.85 |
| scan110 | 3.80 | N/A | 4.60 | 5.73 | **2.28** | 7.62 | 3.89 |
| scan114 | 4.67 | 1.10 | 1.38 | **1.20** | 1.25 | 6.27 | 1.90 |
| scan118 | 4.51 | 2.72 | 2.57 | 5.64 | **1.44** | 7.59 | 3.12 |
| scan122 | 4.35 | 1.64 | 1.76 | 6.20 | **1.45** | 6.47 | 3.84 |
| mean | 4.80 | 2.56 | 2.84 | 4.21 | **1.86** | 6.47 | 3.68 |

Table 7: **Evaluation Results on the DTU Dataset with 3 Input Views.** Note the COLMAP fails on scan110 so we take the average over the remaining 14 scenes. We find that without geometric cues, neither Grids nor MLP works well with only 3 input views. When incorporating the monocular geometric cues, the results for both representations are significantly improved. Interestingly, the grid-based representations perform inferior to a single MLP as they are updated only locally and do not have an inductive smoothness bias compared to a monolithic MLP representation.

## 3.4 DTU

**Geometry.** We show per-scene quantitative results on the DTU dataset with 3 input-views in Table 7 and more qualitative results in Fig. 13. We find that without the monocular geometric cues, both MLP and Multi-Res. Grids fail to produce satisfying reconstructions, while with our monocular cues, both methods are improved and are able to reconstruct high-quality meshes. We further show more visualizations on the DTU dataset using all input views in Fig. 15. Compared to state-of-the-art methods, our approach with multi-resolution feature grids produces more accurate reconstructions.

**Novel View Synthesis.** We further compare our novel view synthesis results on the DTU dataset with three input views. As shown in Table 8 and Fig. 14, using monocular geometric cues improves novel view synthesis results significantly.

**Weight Annealing.** As the monocular depth and normal predictor is not perfect, we exponentially anneal the loss weight for the monocular depth consistency and normal consistency loss, $\lambda_2$ and $\lambda_3$, to 0 during the first 200 epochs of optimization. Qualitative comparison in Fig. 16 verifies the importance of weight annealing.

| | PSNR |
|---|---|
| MLP [28] | 17.65 |
| MLP w/ cues | **23.64** |

Table 8: **Novel view synthesis results on DTU (3 Views).**

**Failure cases.** We show a failure case on DTU with 3 input views in Fig. 17. The reconstructed mesh duplicates the object in front of each camera frustum. One reason is that the monocular depth cues that we use are only up to scale so they do not guarantee multi-view consistency. Therefore, the optimization is still underconstrained since the input RGB images and monocular cues can be explained by individual objects in front of the image plane. One possible solution would be incorporating explicit multi-view constraints such as using sparse point clouds from COLMAP [21] as an additional supervision [5].

## 4 Societal Impact

Our method can faithfully reconstruct a 3D scene which can be used for application ranging from virtual reality to robotics. However, it can also have potential negative societal impact. First, our method relies on a general purpose monocular geometric predictor that needs to be trained on large

amounts of data and with large computational resources, which potentially has a negative impact on global climate change. Second, accurate reconstruction of a scene may raise privacy concerns that need to be addressed carefully. Finally, accurate geometry reconstructed by our method can potentially be used for malicious purposes.

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

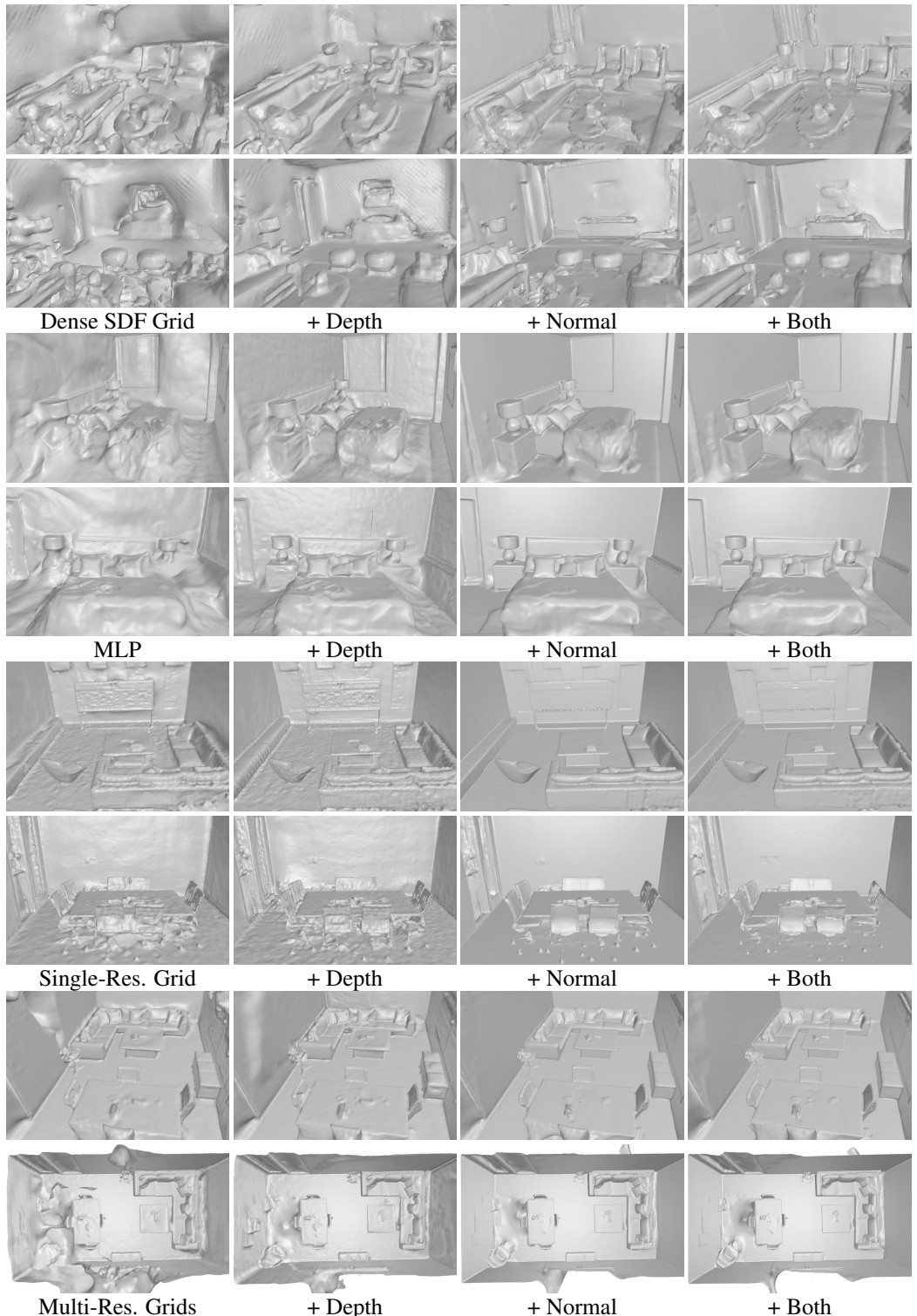

Figure 4: **Ablation of Monocular Geometric Cues on the Replica Dataset.** Monocular geometric cues significantly improve reconstruction quality for all architectures. With monocular depth cues, the recovered geometry contains more details and a better overall structure. Similarly, with our normal cues, missing details are added and the results become smoother. Using both cues leads to the best performance. Zoom in for details.

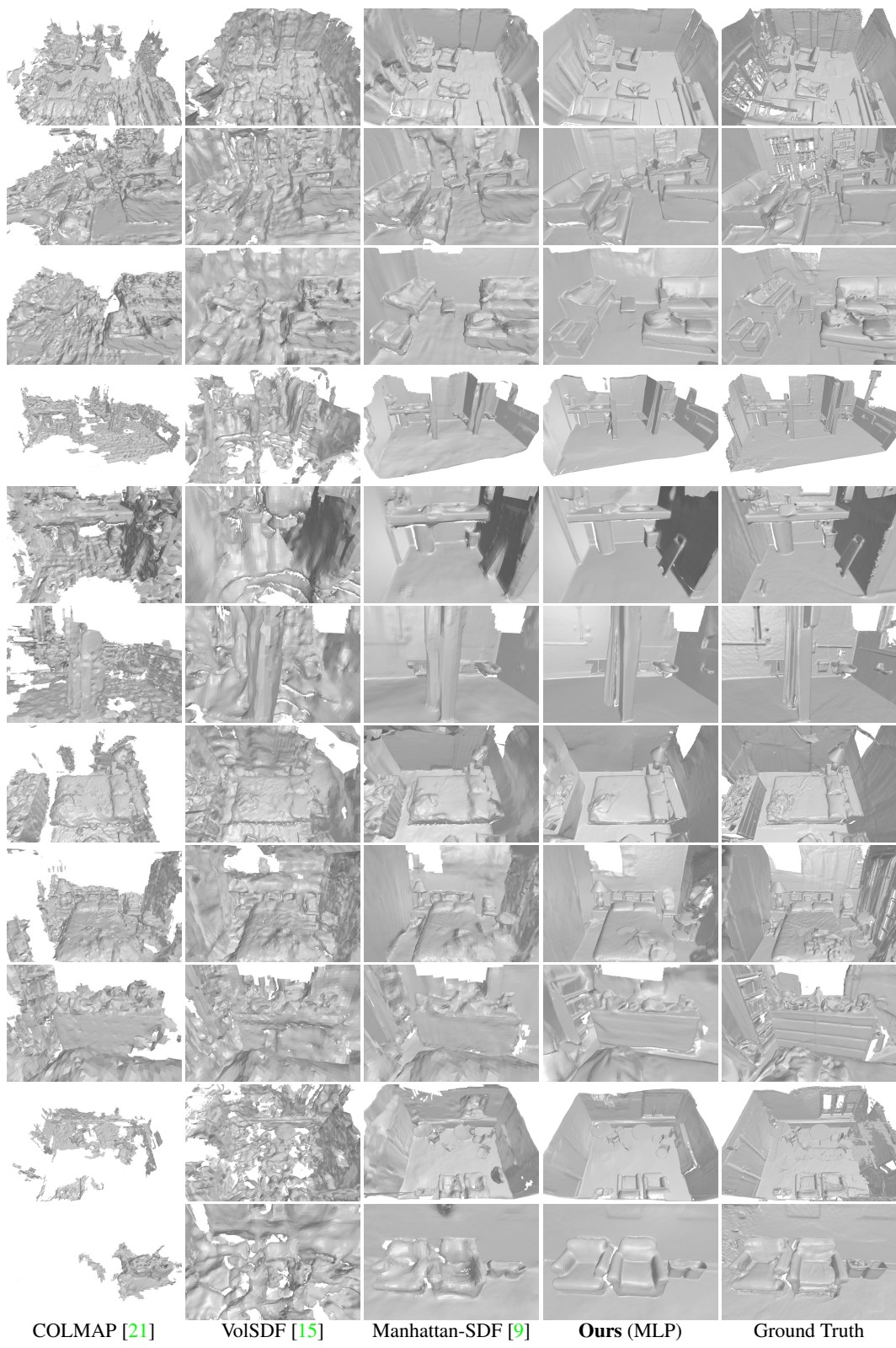

COLMAP [21]     VolSDF [15]     Manhattan-SDF [9]     **Ours** (MLP)     Ground Truth

Figure 5: **Qualitative Comparison on ScanNet.** We show different views for each scene. Our method leads to better results containing smooth surfaces and detailed reconstructions compared against state-of-the-art neural implicit methods.

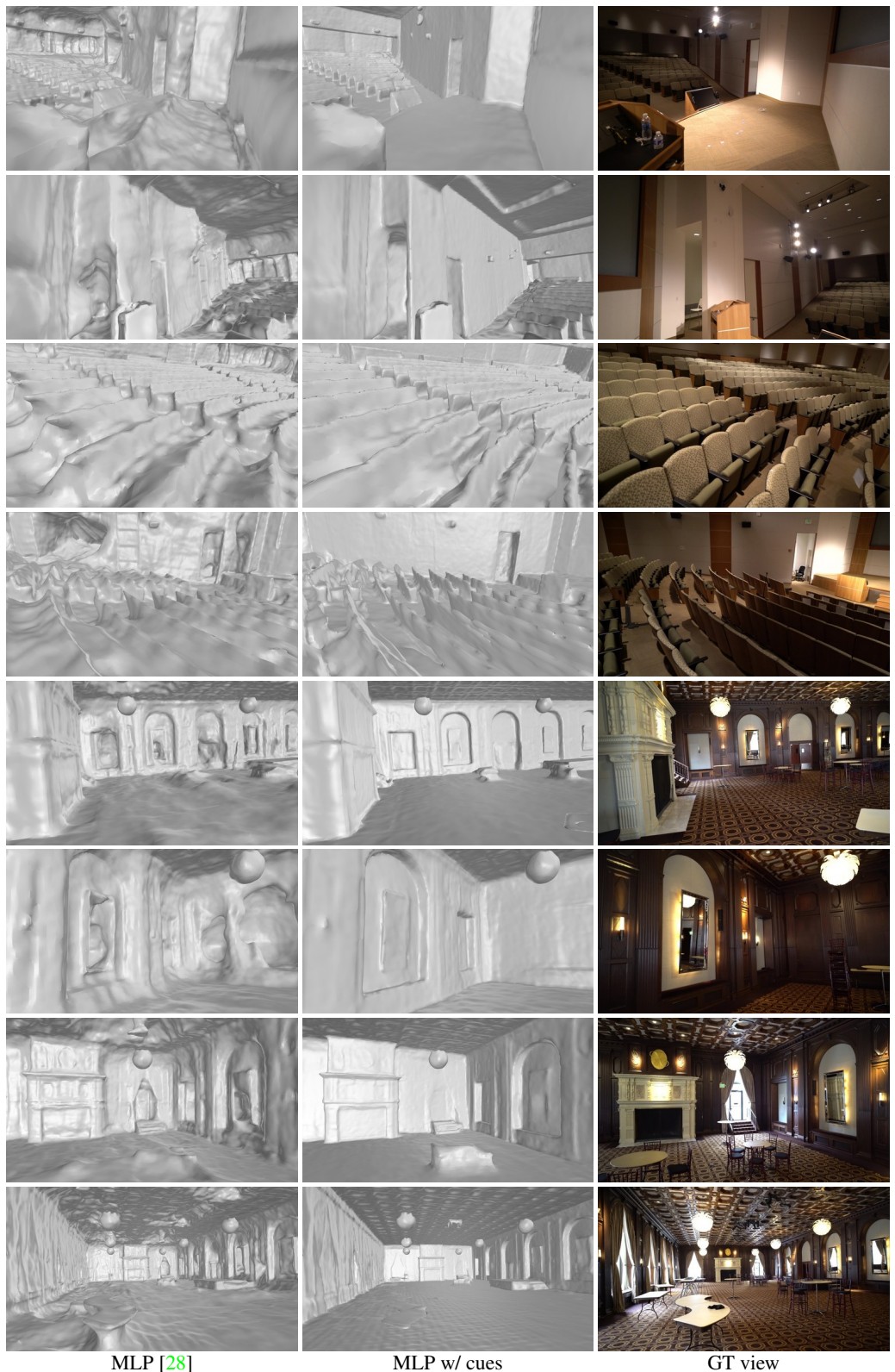

|   |   |   |
|:-:|:-:|:-:|
| MLP [28] | MLP w/ cues | GT view |

Figure 6: **Qualitative Comparison on Tanks & Temples.** We use a single MLP as the scene geometry representation [28] and compare the reconstruction when using monocular cues or not on Auditorium and Ballroom.

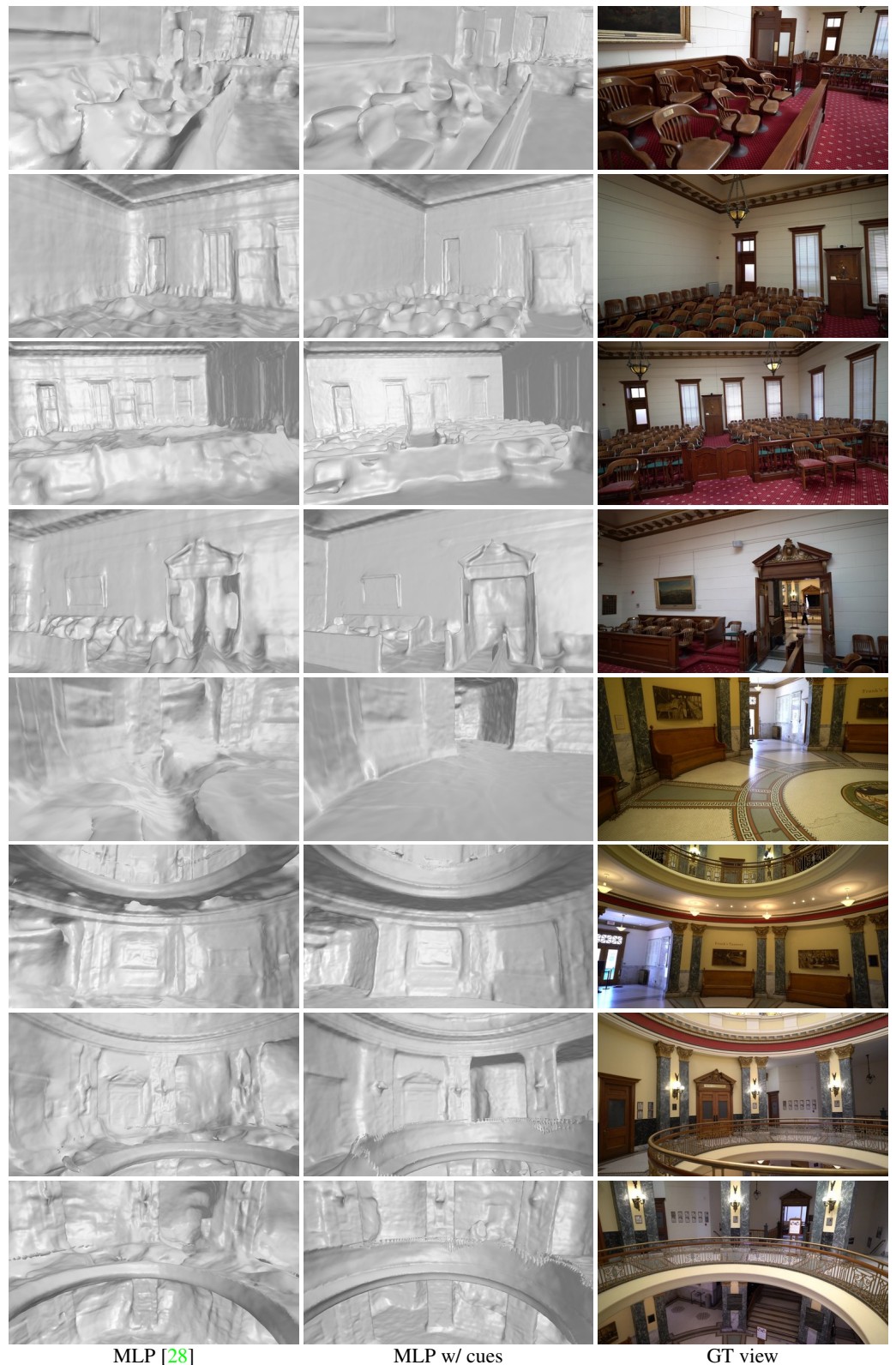

| MLP [28] | MLP w/ cues | GT view |

Figure 7: **Qualitative Comparison on Tanks & Temples Dataset.** We use a single MLP as the scene geometry representation [28] and compare the reconstruction quality when using monocular cues or not on Courtroom and Museum.

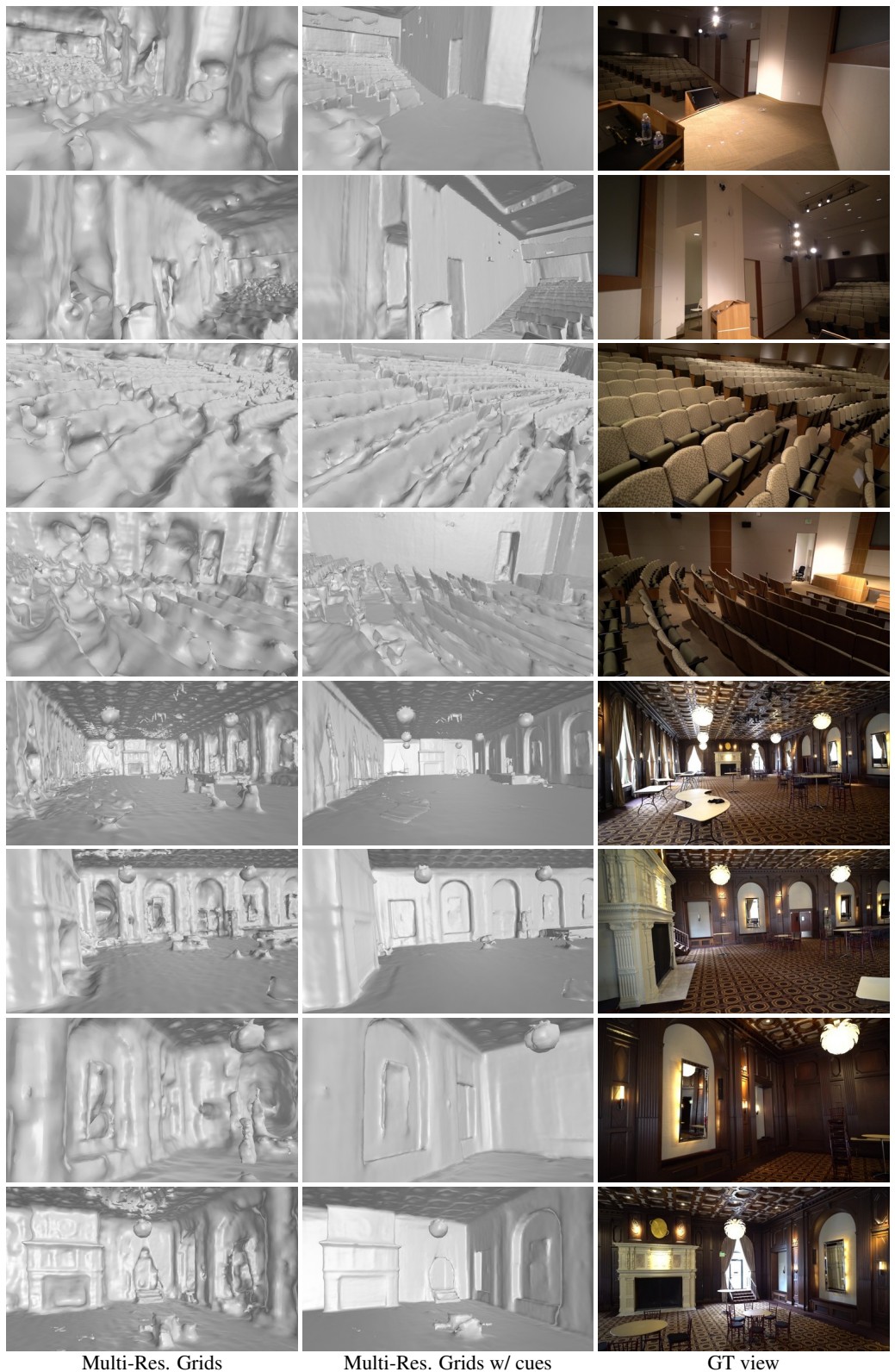

Multi-Res. Grids        Multi-Res. Grids w/ cues        GT view

Figure 8: **Qualitative Comparison on Tanks & Temples.** We use Multi-Res. Grids as the scene geometry representation and compare the reconstruction when using monocular cues or not on Auditorium and Ballroom.

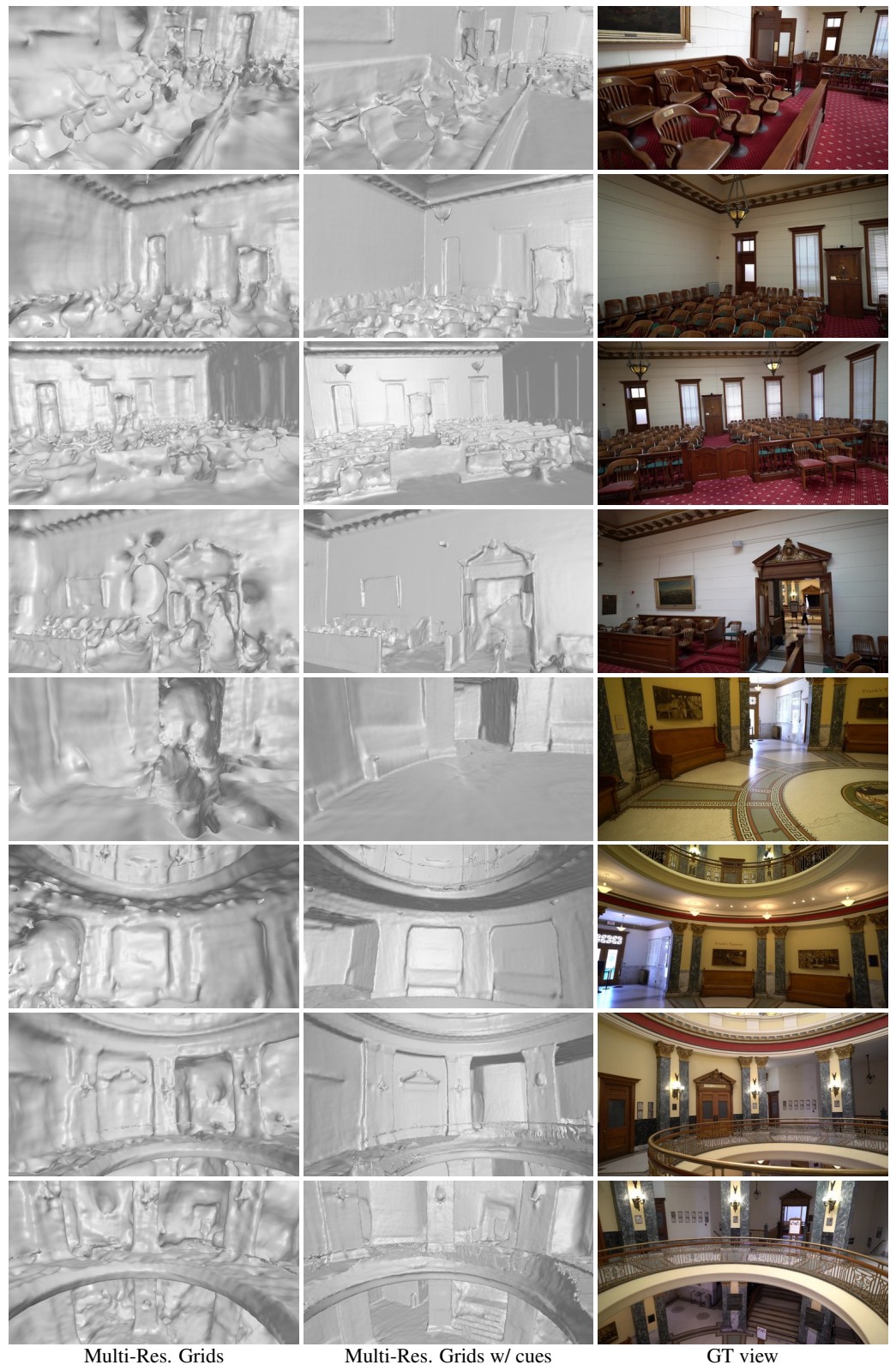

| Multi-Res. Grids | Multi-Res. Grids w/ cues | GT view |

Figure 9: **Qualitative Comparison on Tanks & Temples.** We use Multi-Res. Grids as the scene geometry representation and compare the reconstruction when using monocular cues or not on Courtroom and Museum.

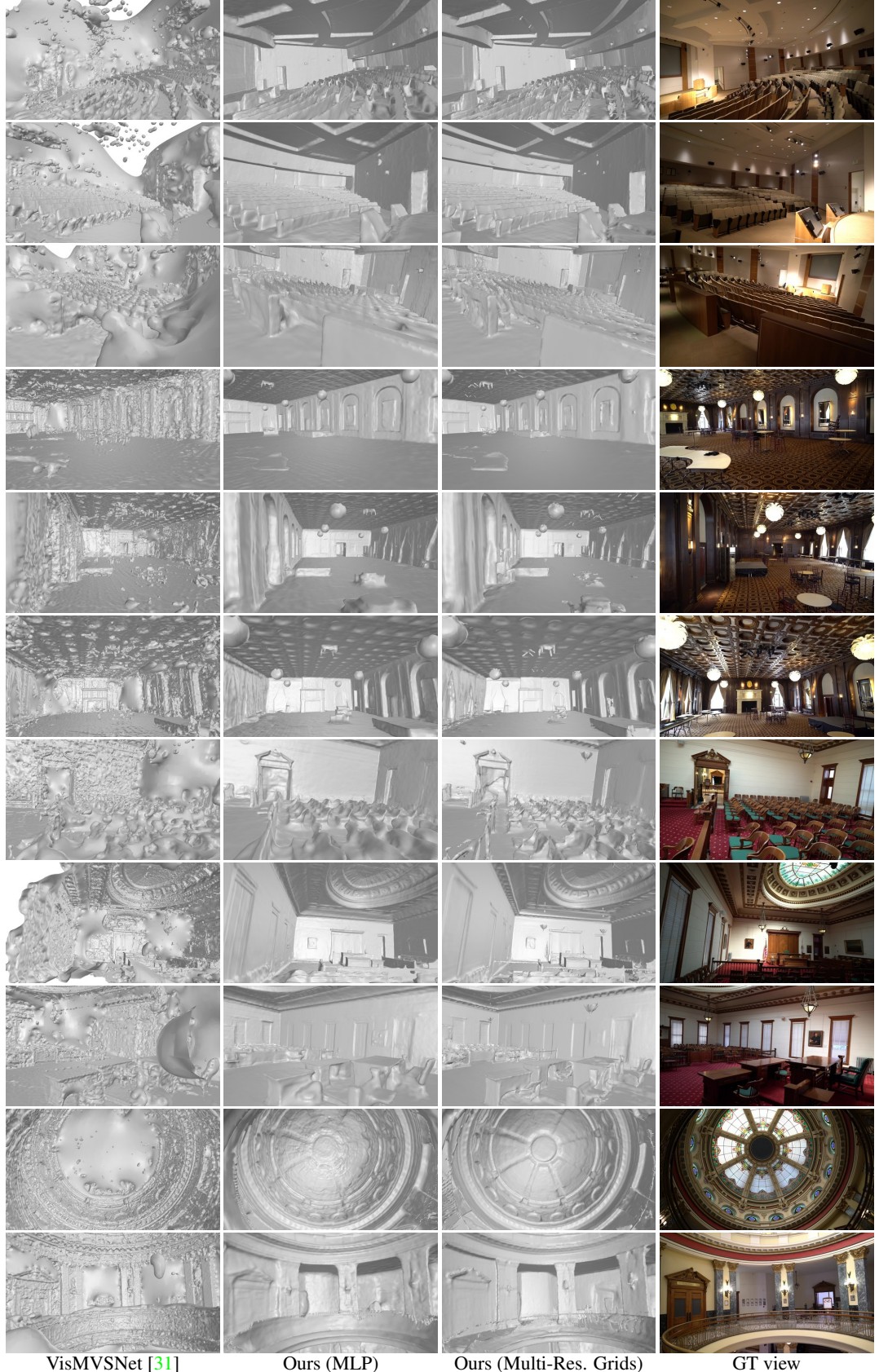

VisMVSNet [31]          Ours (MLP)          Ours (Multi-Res. Grids)          GT view

Figure 10: **Qualitative Comparison on Tanks & Temples.**

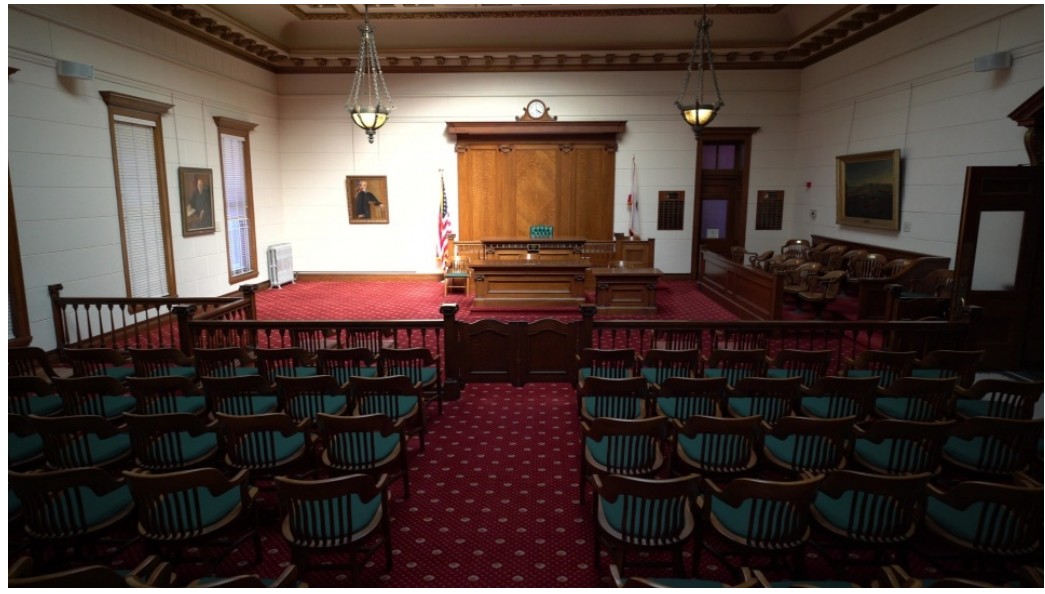

(a) RGB Image.

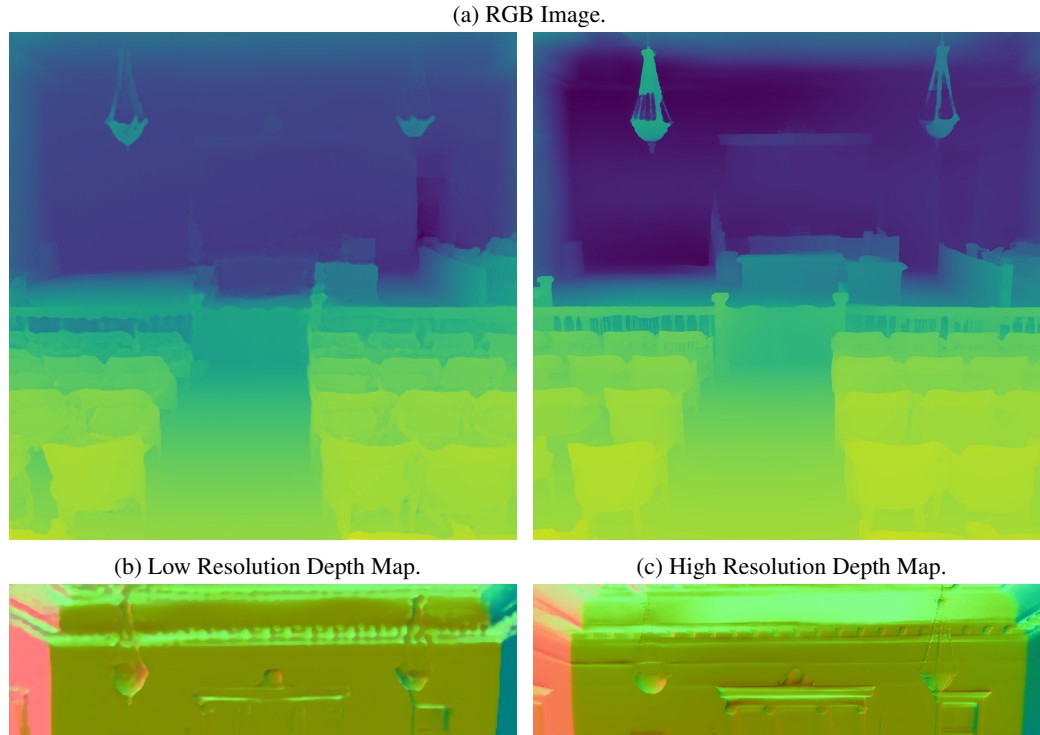

(b) Low Resolution Depth Map.

(c) High Resolution Depth Map.

(d) Low Resolution Normal Map.

(e) High Resolution Normal Map.

Figure 11: **Visual Comparison of Different Resolution Monocular Cues.**

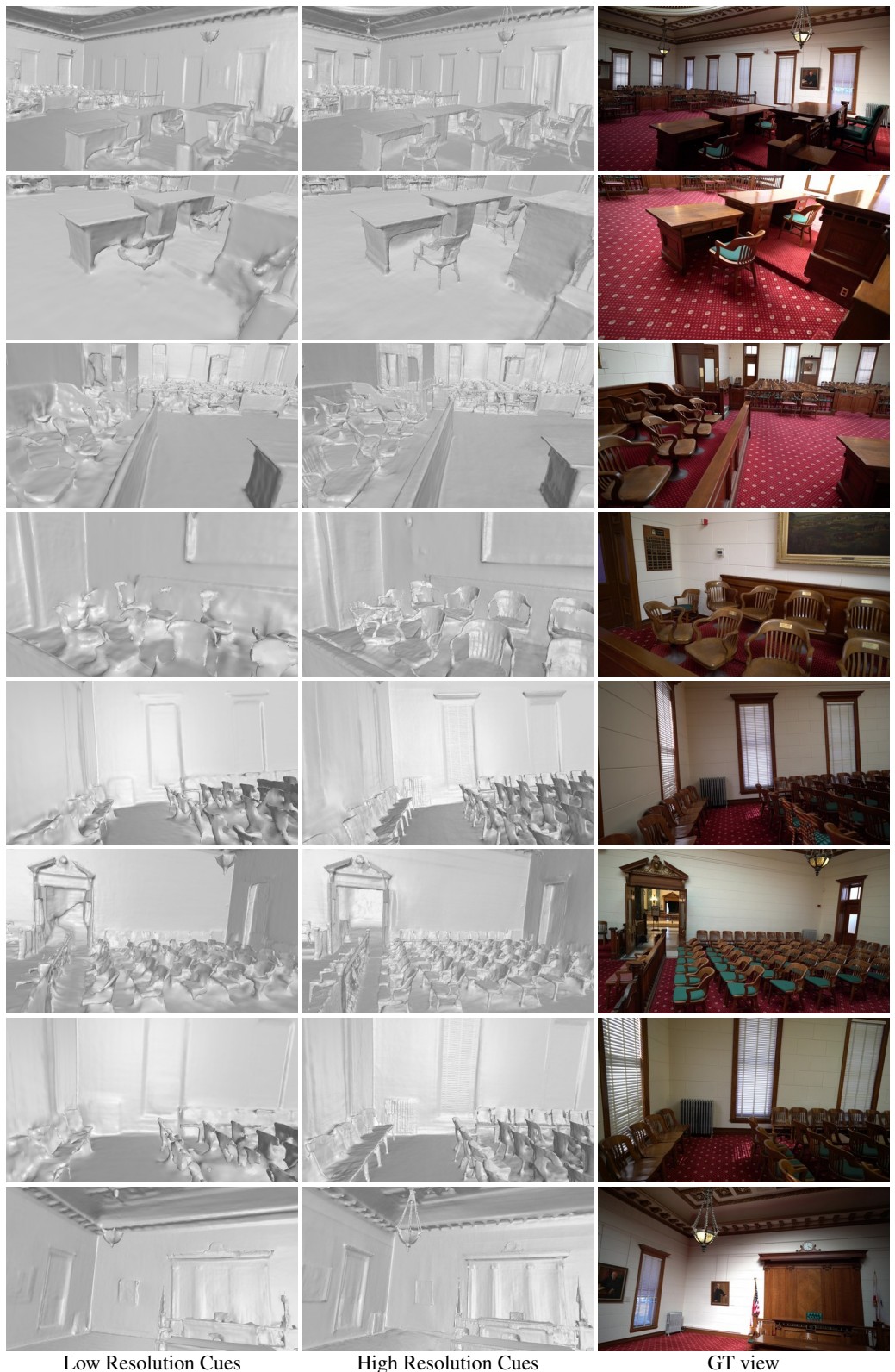

| Low Resolution Cues | High Resolution Cues | GT view |

Figure 12: **Qualitative Comparison of Low Resolution Cues and High Resolution cues on Tanks & Temples.** We use Multi-Res. Grids as the scene geometry representation and compare the reconstruction when using different resolution of monocular cues.

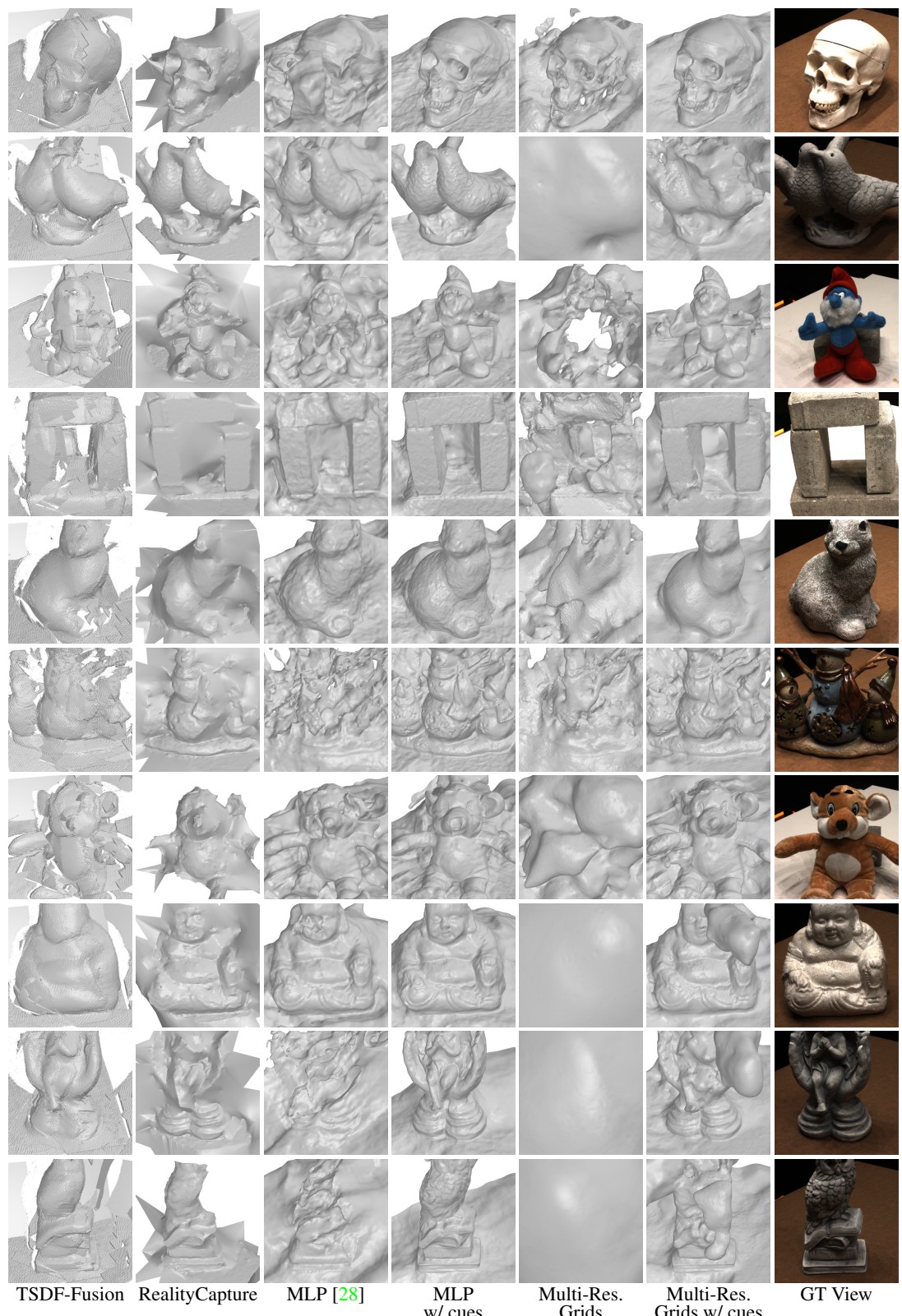

| TSDF-Fusion | RealityCapture | MLP [28] | MLP w/ cues | Multi-Res. Grids | Multi-Res. Grids w/ cues | GT View |
|---|---|---|---|---|---|---|

Figure 13: **Qualitative Comparison on the DTU Dataset with 3 Input Views.** Adding monocular geometric cues improves 3D reconstruction quality for both MLP and Multi-Res. Grids. We show a failure case on the last row.

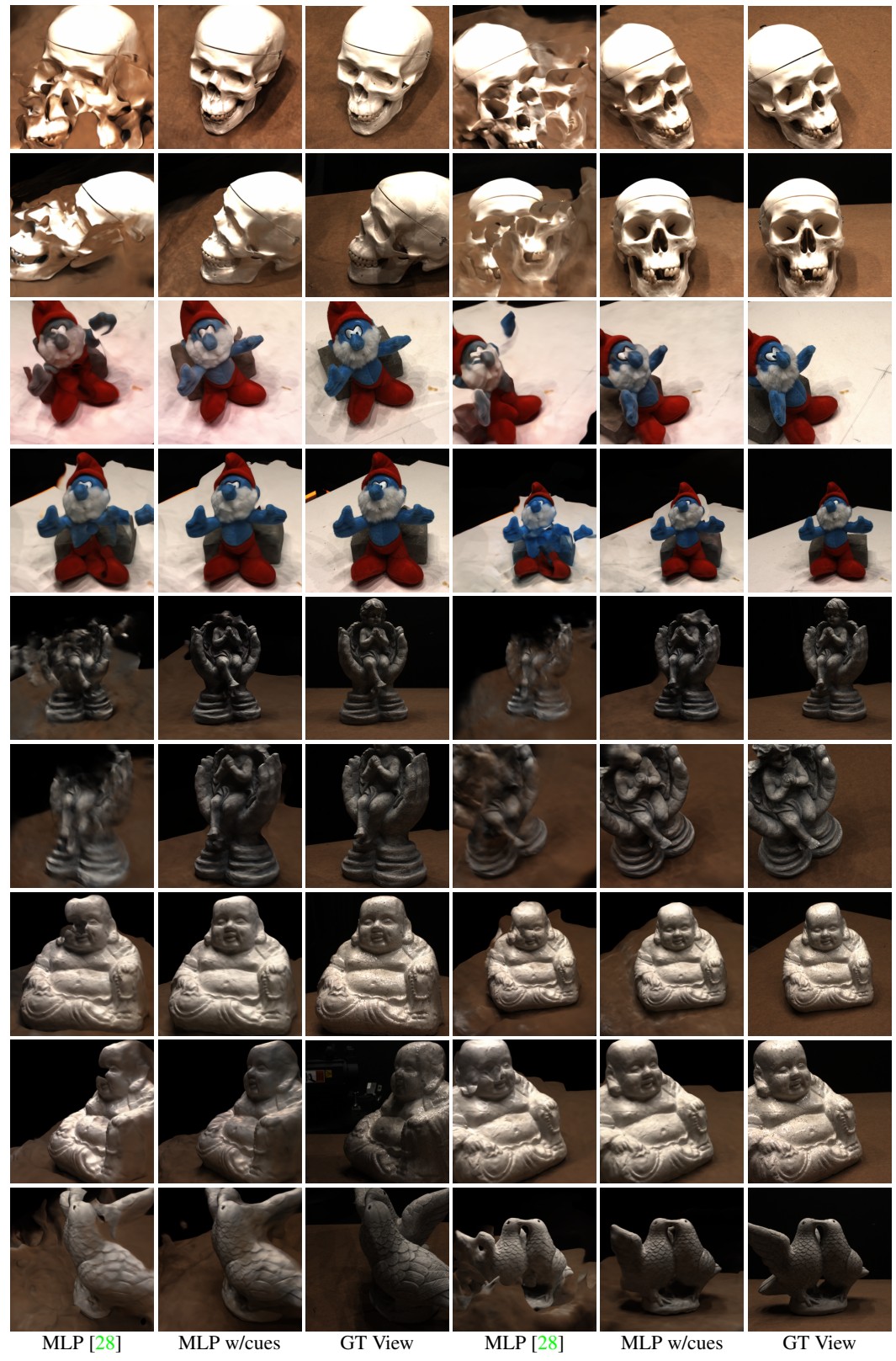

| MLP [28] | MLP w/cues | GT View | MLP [28] | MLP w/cues | GT View |

Figure 14: **Qualitative Comparison of Novel View Synthesis on the DTU Dataset with 3 Input Views.** Adding monocular geometric cues improves novel view synthesis quality.

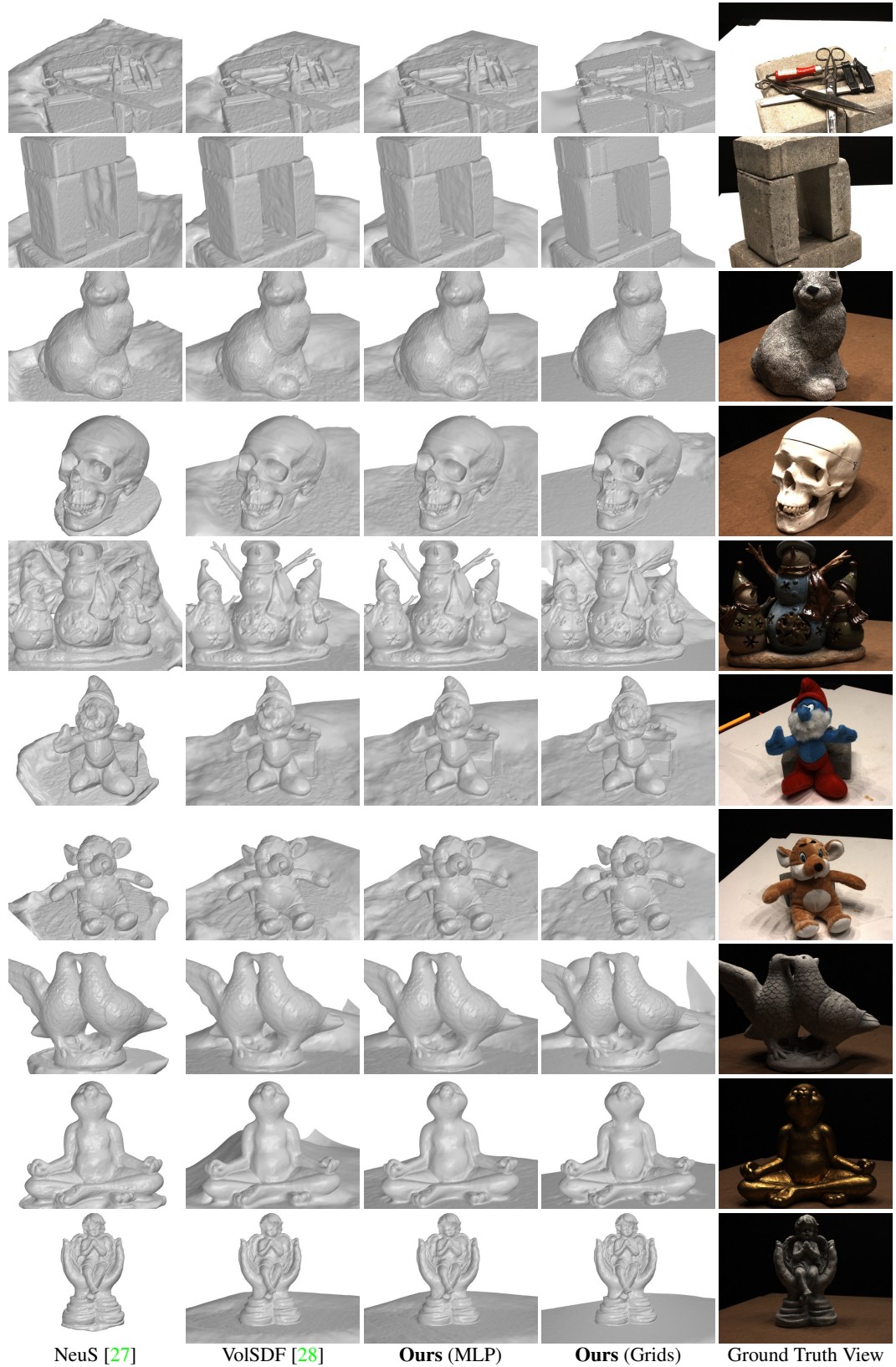

| NeuS [27] | VolSDF [28] | **Ours** (MLP) | **Ours** (Grids) | Ground Truth View |

Figure 15: **Qualitative Comparison on DTU Dataset with all input views.** Our approach with MLP achieves similar results with previous method, while our method with Multi-Res. Fea. Grids reconstruct more detailed surface.

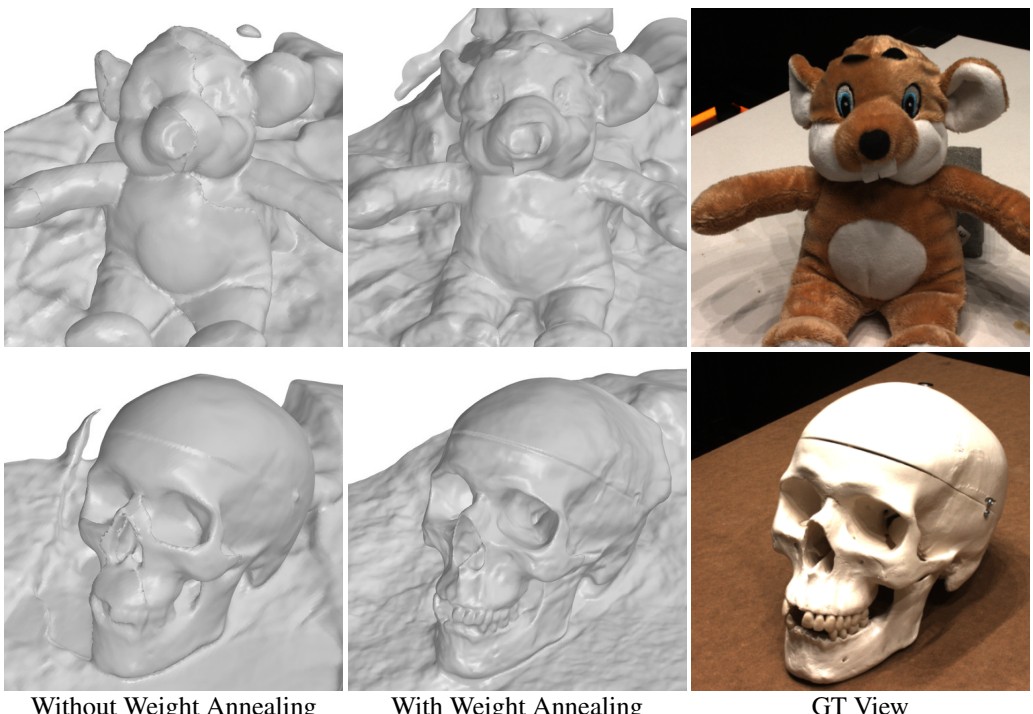

| Without Weight Annealing | With Weight Annealing | GT View |

Figure 16: **Ablation of Weight Annealing on the DTU Dataset with 3 Input Views.** Using weight schedule improves reconstruction quality.

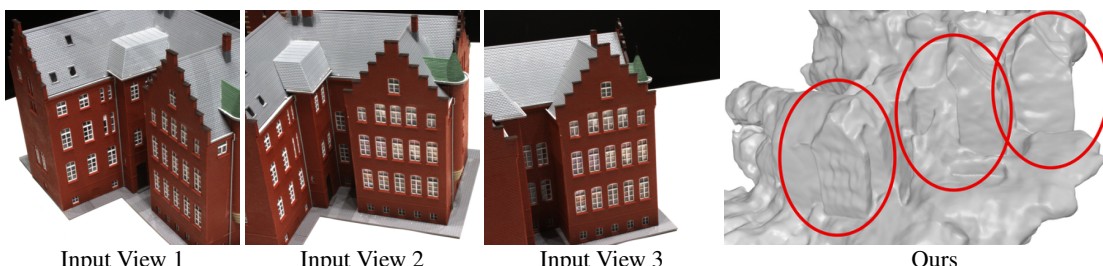

| Input View 1 | Input View 2 | Input View 3 | Ours |

Figure 17: **Failure Case on DTU Dataset with 3 Input Views.** The reconstructed mesh duplicate the object in front of each camera frustum.