# OpenReview forum: "MonoSDF: Exploring Monocular Geometric Cues for Neural Implicit Surface Reconstruction"
_NeurIPS.cc/2022/Conference — NeurIPS 2022 Accept_

### Official Review · Reviewer_grib · 2022-07-06

**Rating:** 8
**Confidence:** 4
**Soundness:** 4 excellent
**Presentation:** 4 excellent
**Contribution:** 4 excellent

**Summary:**

This paper presents a framework to utilize monocular geometric cues to improve multi-view 3D reconstruction quality, efficiency, and scalability for neural implicit surface models. A systematic comparison and detailed analysis of design choices of neural implicit surface representations including vanilla MLP and grid-based approaches has been presented. Among these representations, a simple MLP architecture performs quite well, which demonstrates that MLPs are able to represent complex scenes.

**Questions:**

One ablation study I want to see in the paper is the number of input multi-view images and monocular geometric cues. Specifically, the monocular geometric cues could help the reconstruction when the input images are sparse. If the input images are dense, the monocular geometric cues might influence the reconstruction quality due to the error contained in the monocular cues. I wonder where is the balance for the input number of images? For a reconstruction problem with different number of input images, how should I know whether the monocular geometric cues could help or not?

**Limitations:**

Except the question listed in the above, I don't have other concerns to this paper.

**Strengths And Weaknesses:**

I really like the idea proposed in this paper. To improve the reconstruction quality with sparse input, shape priors should be added. There are several ways to construct the priors. One solution is to construct the parametric model for some special types like face and body. This paper explores another way with the help of depth estimation from single image. Although the estimated depth and normal may contains noises or with wrong scales, the proposed method well handles these issues.

---

> ### Author Response · Authors · 2022-08-02
> **Response to Reviewer grib**
>
> Thank you for your recognition and your time. We are very glad that you consider this paper a technically strong paper with novel ideas and excellent impacts, evaluation, resources. We address your remaining comments below.
>
> **Ablation on the number of input images and monocular geometric cues.**
>
> We ran the experiments with a different number of input images and monocular geometric cues. Please refer to https://imgur.com/a/jqkYC1w for a comparison. We found that adding the monocular geometric cues leads to consistent improvements. We will add these results to our revised paper.

---

> ### Author Response · Authors · 2022-08-08
> **Please let us know if your concerns have been addressed**
>
>
> Dear Reviewer grib,
>
> Thank you again for your review. We hope that our rebuttal could address your questions and concerns. As the discussion phase is nearing its end, we would be grateful to hear your feedback and wondered if you might still have any concerns we could address.
>
> Thank you for your time.

---

> ### Author Response · Authors · 2022-08-09
> **More questions?**
>
>
> Dear Reviewer grib,
>
> Thank you again for your review. We hope that our rebuttal could address your questions. As the discussion phase is only 4 hours left, we wondered if you might still have any concerns we could address.
>
> Thank you for your time.

---

### Official Review · Reviewer_c8RS · 2022-07-09

**Rating:** 7
**Confidence:** 4
**Soundness:** 4 excellent
**Presentation:** 4 excellent
**Contribution:** 3 good

**Summary:**

In this work, the authors proposed a novel and powerful geometric representation using neural implicit function. Previous neural implicit functions are trained purely on RGB reconstruction loss and have difficulty in representing more complicated geometry. In this work, the authors try to address this problem from two directions: first, they propose a novel depth and normal cues that significantly improves the quality of the reconstruction. Secondly, they explored different representation functions, including dense SDF grid, simple MLP, feature grid + MLP and multi-resolution feature grid + MLP. Both of these changes significantly improve the quality of reconstructed geometry.

**Questions:**

I only have a few minor questions for the authors:

In Table 1, in addition to network type, it is also useful to report the number of parameters of each network. Normally, in neural representation, larger network parameters often lead to a better representation power, so this information is important to understand whether the improvement comes from network architecture itself, or simply a larger set of parameters. Moreover, it would be better to compare networks with similar numbers of parameters in Table 1.

The proposed neural representation also recovered the texture of the surface (color). I think it is also important to visualize them to understand the accuracy of texture recovery, even though they are not the main task of this work. It is hard for me to tell whether the network can also represent texture, or the color prediction network is simply to help to train the SDF.

I also have one minor suggestion to this work:

In Figure 3, 4, it would be better to show actual image view or ground truth rendered image, as it is hard to tell whether some small reconstructed geometry are correct or not.



**Limitations:**

No as far as I know.

**Strengths And Weaknesses:**

This is a high quality work. The idea of using depth and normal cues are simple and effective. The proposed multi-resolution feature grid + MLP representation is also novel and effectively improves the reconstruction quality over MLP solution. The experimental results are thorough  and the paper is well written and easy to follow.

I don’t find any particular negative point.

---

> ### Author Response · Authors · 2022-08-02
> **Response to Reviewer c8RS**
>
> Thank you for your insightful comments. We appreciate that you find that our approach is simple and effective, our experimental results are thorough, and our paper is well written. We now address the remaining comments in the following.
>
> **Number of parameters of each network.**
>
> We list the number of parameters of each network used in the following table::
>
> |Method|MLP| Dense SDF Grid | Single-res. Feature Grid | Multi-res. Feature Grid|
> |-|-|-|-|-|
> |Num. Params | 0.7M | 6.4M | 33.8M | 12.8M|
>
> As reviewer D9of asked for a more detailed comparison of different architectures, we ran additional experiments with different architecture configurations. More specifically, we consider MLPs with varying numbers of layers, and Multi-res. Feature Grids with varying sizes of hash tables, to evaluate the performance under different model capacities.
>
> We list the number of learnable parameters under different architecture configurations in the table below and also show their performance over the optimization process in https://imgur.com/a/btDDxEC.
>
> |Model configuration | Num. Params|
> |-|-|
> |MLP (2 layers) | 0.15M |
> |MLP (4 layers) | 0.26M |
> |MLP (8 layers) | 0.63M |
> |MLP (12 layers) | 0.8M |
> |Multi-res. Feature Grids (hash table size $2^{13}$)| 0.41M |
> |Multi-res. Feature Grids (hash table size $2^{15}$)| 1.1M |
> |Multi-res. Feature Grids (hash table size $2^{17}$)| 3.67M |
> |Multi-res. Feature Grids (hash table size $2^{19}$)| 12.67M |
>
> Our experiments show that using monocular geometric cues improves reconstruction quality and convergence speed independent of the network configuration, which supports our claims in the paper. We will add this experiment and the results to the revised version of the paper.
>
> **Visualizing the texture of the surface.**
>
> Please refer to https://imgur.com/a/1dYdvXE for comparing novel view synthesis results on the DTU dataset with three input views. We further a provide quantitative comparison as follows:
>
> |Method|MLP| MLP w/ cues|
> |-|-|-|
> |PSNR | 17.65 | **23.64** |
>
> Using monocular geometric cues improves novel view synthesis results significantly. We will add both qualitative and quantitative results to our revised paper.
>
> **Showing actual image views or ground truth rendered images.**
>
> Thanks for the great suggestion. We will add the ground truth images to our revised paper. We also provide an updated comparison (https://imgur.com/a/19oXNTJ and https://imgur.com/a/LLtMc4E) which includes the ground truth mesh for reference.

---

> > ### Comment · Reviewer_c8RS · 2022-08-08
> > **Feedbacks**
> >
> > Thanks for sharing this additional results.
> >
> > It is pretty clear now the proposed network out-performs previous ones even with similar number of parameters (I am comparing MLP 12 layers v.s. multi-res 2^15). The visual results of rendered RGB also looks reasonable, and would be nice to include more discussion about this in the revision.
> >
> > I don't have any further concerns and will keep my rating as acceptance.

---

> ### Author Response · Authors · 2022-08-08
> **Please let us know if your concerns have been addressed**
>
> Dear Reviewer c8RS,
>
> Thank you again for your review. We hope that our rebuttal could address your questions and concerns. As the discussion phase is nearing its end, we would be grateful to hear your feedback and wondered if you might still have any concerns we could address.
>
> Thank you for your time.

---

### Official Review · Reviewer_Ak1a · 2022-07-11

**Rating:** 5
**Confidence:** 3
**Soundness:** 2 fair
**Presentation:** 2 fair
**Contribution:** 2 fair

**Summary:**

This paper proposed an approach to address 3D reconstruction from multi-view images. The paper is built upon several milestone papers/techniques and the results presented are relatively better. It started off from signed distance function (SDF), and volume rendering of implicit surfaces. The proposal is the incorporation of two losses, i.e. depth and normal consistency estimated from individual images. The estimation, or "ground truth" for supervision, is from a pre-trained Omnidata model [14]. The paper also explored several architectures.

**Questions:**

1. I'm wondering what is the performance comparison between the proposed monocular priors vs self-supervised depth+normal estimation?

**Limitations:**

The paper has addressed the limitations of the existing model it used for depth estimation. However, it does not address whether Omnidata is the best prior to use compared to other models. It also does not address other monocular cues to explore.

**Strengths And Weaknesses:**

*Strength*
1. The paper is easy to read and the main idea is clearly delivered. In essence, the paper took an off-the-shelf depth estimator to serve as a strong prior.
2. The proposed approach is robust across different numbers of images. The method can not only be applied to single objects, but also large scale scenes.

*Weakness*
1. The novelty of this paper is insignificant. It pieces together several techniques (SDF, VolSDF, Omnidata etc). And many of them are well established. I think the paper can potentially compensate this weakness by addressing the points below:
1.1 The exploration of the "monocular geometric cues" is also relatively weak. It is curious to see how different depth estimators may affect the improvement when incorporating the additional prior.
1.2 It is also peculiar that this paper chose depth and normal as the only two monocular cues to experiment. The Omnidata is able to generate high quality ground truth for 19 more more tasks.

2. The color aspect of the reconstruction is never demonstrated or explained. The results are mostly focused on the geometric reconstruction but not the color appearance or rendered 2D images.

3. The experiments for the architectures are confusing. The paper first showed results for comparing different architectures without monocular cues and arrived at the conclusion that the best model is Multi-Res. Fea. Grids. Then after adding the proposed monocular cues, the paper concluded that MLP is the best model. This behavior is not well explained.

---

> ### Author Response · Authors · 2022-08-02
> **Response to Reviewer Ak1a**
>
> Thank you for your valuable comments and constructive feedback. We address your concerns in the following.
>
> **The exploration of the "monocular geometric cues" is relatively weak. It is curious to see how different depth estimators may affect the improvement when incorporating the additional prior.**
>
> Thanks for the question. We ran additional experiments to compare different depth estimators, yielding the following results:
>
> |Method|MLP| w/ MiDaS depth [1] | w/ LeReS depth [2]| w/ Omnidata depth|
> |-|-|-|-|-|
> |F-score | 64.2 | 68.6 | 72.6 | 86.7|
>
> As shown in the table, adding monocular depth cues improves our performance over a single MLP, independently from the depth predictors we use. Unsurprisingly, better depth predictors lead to better performance, with the state-of-the-art Omnidata model giving the best results. We thus believe that the development of better depth cues will further improve the performance of our approach.
>
> Moreover, we tested our model with different monocular normal predictors:
>
> |Method|MLP| w/ Tilted normal [3] | w/ Omnidata normal|
> |-|-|-|-|
> |F-score | 64.2 | 78.3 | 92.2 |
>
>
> We find that both monocular normal predictors improve the results, and similarly to our observations above, using normals predicted by the state-of-the-art Omnidata model leads to the best performance. We will add these results to the final paper.
>
> **It is also peculiar that this paper chose depth and normal as the only two monocular cues to experiment. The Omnidata is able to generate high quality ground truth for 19 more more tasks.**
>
> We use depth and normal cues as they are directly related to geometry and naturally integrate into the volume rendering formulation (see L166-L178 in the main paper). Our experiments show that using these two cues already significantly improves performance, even compared to Manhattan-SDF, which uses depth, planarity, and semantic cues. We agree that exploring other monocular cues such as occlusion edges and curvature is an interesting future direction.
>
> **Appearance aspect of the reconstruction.**
>
> Thanks for the suggestion! Please refer to https://imgur.com/a/1dYdvXE for comparing novel view synthesis results on the DTU dataset with three input views. We further provide quantitative results:
>
>
> |Method|MLP| MLP w/ cues|
> |-|-|-|
> |PSNR | 17.65 | **23.64** |
>
> Using monocular geometric cues improves novel view synthesis results significantly. We will add both qualitative and quantitative results to our revised paper.
>
>
> **Experiments for the architectures are confusing.**
>
> We first evaluate the different architectures without the cues to establish baseline results in order to measure the improvements obtained with the cues. Without using monocular cues, we find that Multi-res. Feature Grids perform better than MLPs, while after using monocular geometric cues, both architectures improve and achieve similar results on the Replica dataset. We agree that this is an interesting finding. We hypothesize that for the MLP, the model capacity is better allocated around surface regions of the 3D geometry if additional prior information is used during optimization, while the Multi-res. Feature Grids have enough model capacity for both cases, optimization with and without prior information. Further, we find that MLPs perform better than Multi-res. Feature Grids in the datasets with noisy observations (e.g., motion blur in image) and noisy camera poses such as ScanNet. Generally, a single MLP is robust to noises but tends to yield smooth surfaces, while Multi-Res. Feature Grids can capture details and converge fast but are less robust to noise and ambiguities in the input images. We will add this discussion to the paper.
>
> **I'm wondering what is the performance comparison between the proposed monocular priors vs self-supervised depth+normal estimation?**
>
> We ran an experiment using a monocular depth map from a pretrained state-of-the-art self-supervised indoor depth estimator [4] in our framework:
>
> |Method|MLP| w/ self-supervised depth [4] | w/ Omnidata depth|
> |-|-|-|-|
> |F-score | 64.2 | 45.6 | 86.7 |
>
> The self-supervised depth estimator degrades performance. We hypothesize that this is due to the weaker performance of the self-supervised model which is also trained with an RGB loss and hence suffers from the under-constrained problem of recovering geometry from multi-view images. We will add these results to the paper.
>
> **References**
>
> [1] Ranftl et al., Towards Robust Monocular Depth Estimation: Mixing Datasets for Zero-Shot Cross-Dataset Transfer, T-PAMI 2022
>
> [2] Yin et al., Learning to Recover 3D Scene Shape from a Single Image, CVPR 2021
>
> [3] Do et al., Surface Normal Estimation of Tilted Images via Spatial Rectifier, ECCV 2020
>
> [4] Li et al., StructDepth: Leveraging the structural regularities for self-supervised indoor depth estimation, ICCV 2021

---

> > ### Comment · Reviewer_Ak1a · 2022-08-09
> > **My concerns were addressed, and I'm willing to increase my score**
> >
> > Thanks to the authors for addressing my concerns. After reading the rebuttal, I still find the experiments on architectural choices distracting to the main contribution of this paper. In essence, the results showed that different architectures respond differently to adding monocular depth and normal cues. If the objective is to maximize the accuracy *with monocular cues*, then the experimentation should be leaned towards that scenario, rather than simply comparing the four architectures for the purpose of constructing a baseline. After all, the architecture is another topic.
> >
> > I'm willing to increase my score as my other concerns have been addressed. My rating is borderline accept as I think the contribution/improvement is largely due to the leverage of pre-trained omnidata model for monocular depth and normal. The paper also has a gap between the title and the content. The title says it's exploring "monocular geometric cues" while the paper is actually resorting to extermal models with monocular depth and normals, which is more like a prior and not what I'm expecting after reading the title.

---

> > > ### Author Response · Authors · 2022-08-09
> > > **Title and content**
> > >
> > > Thank you very much for your reply and for increasing your score. We are happy to change the title if the reviewers and AC recommend this.

---

> ### Author Response · Authors · 2022-08-08
> **Please let us know if your concerns have been addressed**
>
> Dear Reviewer Ak1a,
>
> Thank you again for your review. We hope that our rebuttal could address your questions and concerns. As the discussion phase is nearing its end, we would be grateful to hear your feedback and wondered if you might still have any concerns we could address.
>
> Thank you for your time.

---

> ### Author Response · Authors · 2022-08-09
> **More questions?**
>
>
> Dear Reviewer Ak1a,
>
> Thank you again for your review. We hope that our rebuttal could address your questions and concerns. As the discussion phase is only 4 hours left, we would be grateful to hear your feedback and wondered if you might still have any concerns we could address.
>
> Thank you for your time.

---

### Official Review · Reviewer_D9of · 2022-07-18

**Rating:** 4
**Confidence:** 5
**Soundness:** 3 good
**Presentation:** 3 good
**Contribution:** 2 fair

**Summary:**

Existing neural fields-based methods fail to reconstruct high-quality surfaces for larger and complex scenes with sparse viewpoints. In this work, the authors inspect the issue as inherent ambiguity in RGB loss which provides insufficient constraints. Inspired by the area of monocular geometry prediction, this paper proposes MonoSDF, which explores the utility of depth and normal cues predicted by general-purpose monocular estimators. Experiments demonstrate the geometric monocular priors significantly improve the performance both for single and multi-object scenes.

**Questions:**

When conducting the comparison between MLP and dense grids, how is it possible to make the comparison fair? For example, MLP-based representation is known to be computationally heavy while having low storage requirements, while dense grids can consume very large memory. It is also hard to measure the “capacity” between the MLP-based model and grid (including multi-resolution grids). Simply comparing the number of learnable parameters might not be enough. Therefore, to conduct a complete exploration of different architecture choices, simple tables (like Table 2) might not be enough. For example, for each choice, a figure is required to see the performance vs.. model capacity vs. trade-offs (spatial, convergence speed).

How robust is the monocular cue supervision to the model? For example, how the results change if there are large errors in the depth or normal predictions

In Eq 13, are w and q estimated for each image separately? In Line 189, it mentioned, “per batch”. Does one batch contain multiple images?


**Limitations:**

The paper discussed the limitations and social impacts.

**Strengths And Weaknesses:**

Strengths:

-	The proposed method, which incorporates monocular predictions to ease geometry learning, is quite clean and easy to apply to any neural implicit method.



Weaknesses:

-	The contribution is weak and relatively incremental. The entire paper can be seen as a combination of two different parts: (1) adding monocular cues as additional supervision to improve MVS; (2) exploring different architecture choices for neural implicit representations.

-	For (1), although the idea is clean and easy to understand (as stated in strengths), it is not new in the scenario of learning neural fields. For example, several papers have already explored using predicted depth and other semantic features in learning NeRF. This work applies a very similar idea to surface reconstruction. However, the methodology did not make any changes (auxiliary loss terms) compared to the case of learning standard NeRF. I would expect different choices or additional analysis due to the task of surface reconstruction instead of simply applying the loss. For example, compared to NVS, will normal cues be more important? How the noise of the prediction affects the geometry quality?

-	For (2) part, it is a bit disconnected from the main story. It is always a nice contribution to conduct a systematic exploration of the best architecture for surface reconstruction. Although the findings from the paper about MLP vs. explicit grids are not surprising, the comparison itself can be a good topic. However, such experiments can be done at any settings with different loss functions.

---

> ### Author Response · Authors · 2022-08-02
> **Response to Reviewer D9of (1)**
>
> Thank you very much for the constructive feedback. We particularly appreciate that you find that our approach is "quite clean and easy to apply to any neural implicit method". We believe that the combination of simplicity (in terms of formulation and implementation), flexibility (in terms of not being tied to a specific scene representation), and state-of-the-art results is a particular strength of our approach. Below, we address the concerns raised in the review.
>
> **The contribution is weak and relatively incremental. The entire paper can be seen as a combination of two different parts: (1) adding monocular cues as additional supervision to improve MVS; (2) exploring different architecture choices for neural implicit representations.**
>
> We respectfully disagree that our contribution "is weak and relatively incremental". To the best of our knowledge, existing approaches that use depth priors, such as DS-NeRF [1], Dense Depth Priors [2], and  Manhattan-SDF [3], obtain these priors from multi-view reconstruction (either sparse point clouds from Structure-from-Motion or dense point clouds from Multi-View Stereo). However, multi-view reconstruction approaches often struggle in texture-less scenes and in scenarios with sparse views, see the table in the next answer block.
>
> In contrast, we use monocular cues which are versatile (i.e., can be extracted from a single image using a feedforward network) and exploit the recognition ability of state-of-the-art deep neural networks as opposed to multi-view reconstruction approaches which utilize photoconsistency cues similar to the NeRF objective itself. Our insight is that photometric consistency cues used by surface reconstruction methods (such as VolSDF) and the recognition cues provided by monocular geometric networks are complementary, see L52-L59 in the paper. We show that such readily available monocular cues can be easily used to significantly increase 3D reconstruction quality, especially in challenging settings such as 3-view DTU and Tanks \& Temples. We are not aware of any prior work based on neural implicit scene representations which yields good results for the advanced split of the Tanks \& Temples dataset.
>
> Finally, using monocular rather than multi-view cues comes with its own challenges, most notably that depth is only defined up to an unknown scaling factor and an unknown shift per depth map and that the cues can be rather inaccurate. As such, the losses used by our approach differ significantly from the losses used by methods which integrate multi-view depth constraints (eg, DS-NeRF), by taking scale-invariance into account and modeling normal consistency.
>
> **For (1), although the idea is clean and easy to understand (as stated in strengths), it is not new in the scenario of learning neural fields. For example, several papers have already explored using predicted depth and other semantic features in learning NeRF. This work applies a very similar idea to surface reconstruction.**
>
> As argued above, our approach uses a fundamentally different source of depth cues (monocular predictions) compared to existing work (multi-view predictions).
> Geometrically, our cues are somewhat weaker as each depth map is defined up to unknown scale and shift values (see also below), thus introducing additional parameters that need to be estimated. Our paper demonstrates that estimating these parameters during optimization is possible, leading to consistently improved 3D reconstruction results when integrating weak monocular cues.
>
> The table below compares our approach, based on geometrically weak monocular cues, to Manhattan-SDF (which uses multi-view depth and semantic cues, as well as normals obtained from manhattan-world assumption) and multiple variants of VolSDF (used as baselines in [3]) for the ScanNet dataset. As can be seen, our monocular cues significantly improve performance compared to using multi-view cues. Please see [3] for a detailed explanation of the baselines.
>
> |Method|Chamfer-L1|F-score|
> |-|-|-|
> |VolSDF|0.267|0.364|
> |VolSDF + Colmap Depth [4]|0.164|0.431|
> |VolSDF + Colmap Depth [4] + Semantic [5]|0.104|0.474|
> |Manhattan-SDF [3] |0.070|0.602|
> |Ours|**0.042**|**0.733**|
>
>
> [1] Deng et al., Depth-supervised NeRF: Fewer Views and Faster Training for Free, CVPR 2022
>
> [2] Roessle et al., Dense Depth Priors for Neural Radiance Fields from Sparse Input Views, CVPR 2022
>
> [3] Guo et al., Neural 3D Scene Reconstruction with the Manhattan-world Assumption, CVPR 2022
>
> [4] Schönberger et al., Pixelwise View Selection for Unstructured Multi-View Stereo, ECCV 2016
>
> [5] Chen et al., Encoder-decoder with atrous separable convolution for semantic image segmentation, ECCV2018

---

> > ### Author Response · Authors · 2022-08-02
> > **Response to Reviewer D9of (2)**
> >
> > **However, the methodology did not make any changes (auxiliary loss terms) compared to the case of learning standard NeRF. I would expect different choices or additional analysis due to the task of surface reconstruction instead of simply applying the loss.**
> >
> > There surely are more complex ways to utilize monocular cues, e.g., to reduce the number of samples by avoiding sampling in empty space. The advantage of using auxiliary loss terms compared to a deeper integration is flexibility, allowing us to use monocular cues independently of the used scene representation. At the same time, this "simple" way of integrating monocular cues already leads to significant and consistent improvements in challenging scenes. We consider the combination of simple and flexible integration and strong results a strength rather than a weakness of our approach.
> >
> > **For example, compared to NVS, will normal cues be more important?**
> >
> > We find that both depth and normal cues are complementary and improve reconstruction results. We achieve the best results with both cues, see table 2 in the main paper. Nevertheless, we indeed found that normal cues lead to relatively more improvements, especially when using an MLP instead of a multi-res. grid representation.
> >
> > **For (2) part, it is a bit disconnected from the main story.**
> >
> > We make the general statement that incorporating monocular cues significantly improves performance. In order to verify this statement, we need to show that the cues improve performance independently of the chosen scene representation.
> > Thus, exploring the impact of the cues on various commonly used representations is a central part of our study. Moreover, a surprising conclusion from our experiments is that otherwise inferior MLP representations are able to attain performance on par with more recent multi-resolution feature grids when exploiting monocular cues, see table 2 in the main paper. We believe that these results are interesting and worth to be shared with the research community.
> >
> >
> > **The findings from the paper about MLP vs. explicit grids are not surprising.**
> >
> > In our experiments on ScanNet, Multi-res. Feature grids lead to noiser reconstruction compared to MLPs due to dataset low image quality (e.g., motion blur) and noisy camera poses. We argue that this was not obvious before our experiments and our results, for the first time, reveal that the MLP architecture is more robust to noisy inputs compared to Multi-res feature grids. Generally, a single MLP is robust to noises but tends to yield smooth surfaces, while Multi-Res. Feature Grids are able to capture details and converge fast but are less robust to noise and ambiguities in the input images. Using monocular cues, however, we surprisingly found that a simple MLP architecture performs best overall, demonstrating that MLPs, in principle, can represent complex scenes while converging more slowly compared to grid-based representations. We believe our findings will provide valuable insights for future work.
> >
> > **More comparison of different architecture configurations.**
> >
> > Thanks for the great suggestion! We agree that exploring different configurations is interesting and thus performed experiments with different architectural configurations. In order to evaluate the performance with different model capacities, we consider MLPs with a different number of layers and multi-resolution feature grids with different sizes of the hash table.
> >
> > We list the number of learnable parameters using different architecture configurations in the table below, and also show their performance over the optimization processes in https://imgur.com/a/btDDxEC.
> > Our experiments show that using monocular geometric cues improves reconstruction quality and convergence speed independent of the network configuration, which is consistent with our findings in the paper. We will add this experiment to the revised version of the paper.
> >
> >
> > |Model configuration | Num. Params|
> > |-|-|
> > |MLP (2 layers) | 0.15M |
> > |MLP (4 layers) | 0.26M |
> > |MLP (8 layers) | 0.63M |
> > |MLP (12 layers) | 0.8M |
> > |Multi-res. Feature Grids (hash table size $2^{13}$)| 0.41M |
> > |Multi-res. Feature Grids (hash table size $2^{15}$)| 1.1M |
> > |Multi-res. Feature Grids (hash table size $2^{17}$)| 3.67M |
> > |Multi-res. Feature Grids (hash table size $2^{19}$)| 12.67M |

---

> > > ### Author Response · Authors · 2022-08-02
> > > **Response to Reviewer D9of (3)**
> > >
> > > **How robust is the monocular cue supervision to the model, e.g., when there are noise or large errors in the monocular predictions**
> > >
> > > Thanks for the question. We did not observe any significant errors in the monocular predictions in the real-world datasets (ranging from object to large-scale indoor scenes). Even for ScanNet dataset where motion blurs or noises are present in the RGB images, the predicted monocular depths and normals are still of high quality. However, we do observe that the model predicts infinite depth in some region of the Replica dataset (e.g. windows or doors with completely black colors). We filter out these images if the maximum depth is ten times larger than the minimum depth in the image.
> > >
> > > To further analyze the robustness of our approach to monocular geometric cues of different levels of quality, we further tested our model with different depth predictors:
> > >
> > > |Method|MLP| w/ MiDaS depth [6] | w/ LeReS depth [7]| w/ Omnidata depth|
> > > |-|-|-|-|-|
> > > |F-score | 64.2 | 68.6 | 72.6 | 86.7|
> > >
> > > For all three methods, adding monocular depth improves performance over a single MLP without cues. Unsurprisingly, better depth predictors lead to better performance, with the state-of-the-art Omnidata model giving the best results. We thus believe that the development of better depth cues will further improve the performance of our approach.
> > >
> > > We also tested our model with different monocular normal predictors and obtain the following results:
> > >
> > >
> > > |Method|MLP| w/ Tilted normal [8]| w/ Omnidata normal|
> > > |-|-|-|-|
> > > |F-score | 64.2 | 78.3 | 92.2 |
> > >
> > > We find that both monocular normal predictors improve the results, and similarly to our observations above, using normals predicted by the state-of-the-art Omnidata model leads to the best performance. We will add this discussion and both experiments to our revised paper.
> > >
> > > **How are $w$ and $q$ estimated?**
> > >
> > > $w$ and $q$ are estimated per image as each depth map is defined up to an unknown scale and shift. In our implementation, we sample one image randomly in each iteration and then sample a batch of rays within the image. Then we estimate $w$ and $q$ using this batch of rays with a least-squares criterion which has closed-form solution, see L26-L36 in the supplementary document. We will make this clear in our revised paper.
> > >
> > > [6] Ranftl et al., Towards Robust Monocular Depth Estimation: Mixing Datasets for Zero-Shot Cross-Dataset Transfer, T-PAMI 2022
> > >
> > > [7] Yin et al., Learning to Recover 3D Scene Shape from a Single Image, CVPR 2021
> > >
> > > [8] Do et al., Surface Normal Estimation of Tilted Images via Spatial Rectifier, ECCV 2020

---

> ### Author Response · Authors · 2022-08-08
> **Please let us know if your concerns have been addressed**
>
> Dear Reviewer D9of,
>
> Thank you again for your review. We hope that our rebuttal could address your questions and concerns. As the discussion phase is nearing its end, we would be grateful to hear your feedback and wondered if you might still have any concerns we could address.
>
> Thank you for your time.

---

> ### Author Response · Authors · 2022-08-09
> **More questions?**
>
> Dear Reviewer D9of,
>
> Thank you again for your review. We hope that our rebuttal could address your questions and concerns. As the discussion phase is only 4 hours left, we would be grateful to hear your feedback and wondered if you might still have any concerns we could address.
>
> Thank you for your time.

---

> > ### Comment · Reviewer_D9of · 2022-08-09
> > **I have read the rebuttal...**
> >
> > Hi, thanks so much for the very long response to my questions.
> > However, I still think the contribution is relatively incremental and weak, and I disagree with your response.
> > I don't think there is a fundamental difference between using additional "cues" either from multi-view reconstruction or monocular predictions. Both have pros and cons, and there are several papers that have already done that.

---

> > > ### Author Response · Authors · 2022-08-09
> > > **More details?**
> > >
> > > Dear Reviewer D9of,
> > >
> > > Thanks for your reply. First, using depth maps from multi-view stereo as additional supervision fails in the indoor scenes with lots of textureless regions, see the table in our response (1). Because multi-view stereo methods use photometric cues to establish correspondences, they struggle in texture-less scenes and scenarios with sparse views. In contrast, monocular depth and normal estimators are trained on large-scale datasets in a supervised manner, and their outputs can serve as strong prior for optimizing neural implicit models.
> > >
> > > Second, to the best of our knowledge, our approach is the first method that utilizes monocular depth and normal cues to neural implicit surface models and has demonstrated significant improvements over baseline methods on several challenging datasets. We are not aware of prior work that studied mono cues in the context of Neural implicit surface reconstruction and would be grateful if you could provide more details regarding prior work that investigated this setting.
> > >
> > > Thank you very much for your time.

---

> > > > ### Comment · Reviewer_D9of · 2022-08-09
> > > > **I am willing to increase score to 4, but...**
> > > >
> > > > Thanks for the response. Regarding the details which have partially solved my questions. I am willing to increase my score.
> > > >
> > > > However, I am still not too convinced by the proposed novelty. Whether the proposed method is incremental or not is not checking if there is anyone working on exactly the same idea or not, otherwise, anyone can find one point which is definitely not investigated by previous papers. I understood the paper wanted to show the importance of "monocular" cues compared to cues in general. Then it should include more experiments to show why it is critical.
> > > >
> > > > Therefore, I still tend to keep my negative rating.
> > > >
> > > > Also, the following paper seems also uses normal supervision from monocular prediction.
> > > > https://arxiv.org/pdf/2206.13597.pdf

---

> > > > > ### Author Response · Authors · 2022-08-09
> > > > > **The paper should be considered as concurrent work**
> > > > >
> > > > > Dear Reviewer D9of,
> > > > >
> > > > > Thank you very much for your reference and for increasing your score.
> > > > >
> > > > > First, please note that the reference paper is submitted to arXiv after the NeurIPS deadline and should be considered as concurrent work. Second, compared to this paper, our approach is not restricted to indoor scenes, and we did experiments on the challenging 3-view setting in the DTU dataset. Our approach is the first method that yields reasonable reconstruction on the large-scale Tanks and Temples dataset. Further, we demonstrated the effectiveness of monocular cues on various neural scene representations, ranging from MLP to multi-res. feature grids.
> > > > >
> > > > > Regarding comparison to other cues, we provided a comparison with Manhattan-SDF, which utilises semantic and multi-view stereo depth maps, and other baselines in our response (1). Our monocular cues significantly improve the reconstruction quality.
> > > > >
> > > > > We believe our approach is well-qualified and brings value to the NeurIPS community, and we would be grateful if you could increase your score to acceptance.
> > > > >
> > > > > Thank you very much for your time.

---

> > > > > > ### Comment · Reviewer_D9of · 2022-08-09
> > > > > > **No, I will keep my negative rating**
> > > > > >
> > > > > > Thanks for the kind response.

---

### Meta-Review · Area_Chair_mwxk · 2022-08-31

**Recommendation:** Accept
**Confidence:** Certain

**Metareview:**

There was a range of reactions to this paper from borderline reject to strong accept.  Although several of the reviewers highlighted that the contribution could be viewed as incremental, it is clearly described, and robust across different types of scenes, and I concur with the three reviewers that give positive ratings.  Therefore I am accepting this paper.

**Award:**

No

---

### Decision · Program_Chairs · 2022-09-14

Accept